

# The osteology and phylogenetic position of the loricatan (Archosauria: Pseudosuchia) *Heptasuchus clarki*, from the ?Mid-Upper Triassic, southeastern Big Horn Mountains, Central Wyoming (USA)

Sterling J. Nesbitt[1], John M. Zawiskie[2,3,†] Robert M. Dawley[4]

[1] Department of Geosciences, Virginia Tech, Blacksburg, VA, USA
[2] Cranbrook Institute of Science, Bloomfield Hills, MI, USA
[3] Department of Geology, Wayne State University, Detroit, MI, USA
[4] Department of Biology, Ursinus College, Collegeville, PA, USA
[†] Deceased author.

## ABSTRACT

Loricatan pseudosuchians (known as "rauisuchians") typically consist of poorly understood fragmentary remains known worldwide from the Middle Triassic to the end of the Triassic Period. Renewed interest and the discovery of more complete specimens recently revolutionized our understanding of the relationships of archosaurs, the origin of Crocodylomorpha, and the paleobiology of these animals. However, there are still few loricatans known from the Middle to early portion of the Late Triassic and the forms that occur during this time are largely known from southern Pangea or Europe. *Heptasuchus clarki* was the first formally recognized North American "rauisuchian" and was collected from a poorly sampled and disparately fossiliferous sequence of Triassic strata in North America. Exposed along the trend of the Casper Arch flanking the southeastern Big Horn Mountains, the type locality of *Heptasuchus clarki* occurs within a sequence of red beds above the Alcova Limestone and Crow Mountain formations within the Chugwater Group. The age of the type locality is poorly constrained to the Middle—early Late Triassic and is likely similar to or just older than that of the Popo Agie Formation assemblage from the western portion of Wyoming. The holotype consists of associated cranial elements found in situ, and the referred specimens consist of crania and postcrania. Thus, about 30% of the osteology of the taxon is preserved. All of the pseudosuchian elements collected at the locality appear to belong to *Heptasuchus clarki* and the taxon is not a chimera as previously hypothesized. *Heptasuchus clarki* is distinct from all other archosaurs by the presence of large, posteriorly directed flanges on the parabasisphenoid and a distinct, orbit-overhanging postfrontal. Our phylogenetic hypothesis posits a sister-taxon relationship between *Heptasuchus clarki* and the Ladinian-aged *Batrachotomus kupferzellensis* from current-day Germany within Loricata. These two taxa share a number of apomorphies from across the skull and their phylogenetic position further supports 'rauisuchian' paraphyly. A minimum of three individuals of *Heptasuchus* are present at the type locality suggesting that a group of individuals died together, similar to other aggregations of loricatans (e.g., *Heptasuchus*, *Batrachotomus*, *Decuriasuchus*, *Postosuchus*).

Corresponding author
Sterling J. Nesbitt, sjn2104@vt.edu

## INTRODUCTION

During the Middle and Late Triassic, a variety of large pseudosuchian archosaur predators appeared across Pangea. These forms included long-snouted phytosaurs (*Stocker & Butler, 2013*), sailed-back poposauroids (*Nesbitt, 2003*, *2005*, *2011*; *Butler et al., 2011*; *Nesbitt, Liu & Li, 2011*), short-faced ornithosuchids (*von Baczko & Ezcurra, 2013*), and quadrupedal, large headed "rauisuchians"—a group that has been traditionally classified together (*Nesbitt et al., 2013*). "Rauisuchians" have been found in nearly every well-sampled Middle to Upper Triassic deposit, but the anatomy and the relationships of these pseudosuchians remains debated (*Gower, 2000*; *Brusatte et al., 2008*, *2010*; *Nesbitt, 2011*; *Nesbitt et al., 2013*). Namely, it is not clear if these "rauisuchians" represent a natural group (traditional hypothesis; *Brusatte et al., 2008*; *2010*), a grade leading to crocodylomorphs (*Nesbitt, 2011*), or a combination of subclades and grades spread across Pseudosuchia (*Nesbitt, 2011*; *Nesbitt et al., 2013*). Luckily, over the past 20 years, huge headway has been made in uncovering their anatomy and relationships through the discovery of new taxa (e.g., *Batrachotomus kupferzellensis*; *Gower, 1999*; *Gower & Schoch, 2009*; *Postosuchus alisonae*, *Peyer et al., 2008*; *Decuriasuchus quartacolonia*, *De França, Ferigolo & Langer, 2011*; *De França et al., 2013*; *Viveron haydeni*, *Lessner et al., 2016*; *Mandasuchus tanyauchen*, *Butler et al., 2018*), or new specimens of previously named taxa (e.g., *Arizonasaurus babbitti*, *Nesbitt, 2003*, *2005*; *Prestosuchus chiniquensis*, *Da Silva et al., 2018*; *Mastrantonio et al., 2019*; *Poposaurus gracilis*, *Schachner et al., 2019*) and revised and detailed descriptions (e.g., *Rauisuchus tiradentes*, *Lautenschlager & Rauhut, 2015*; *Postosuchus kirkpatricki*, *Weinbaum, 2011*, *2013*; *Luperosuchus fractus*, *Nesbitt & Desojo, 2017*; *Prestosuchus chiniquensis*, *Desojo, Baczko & Rauhut, 2020*; *Ticinosuchus ferox*, *Lautenschlager & Desojo, 2011*).

"Rauisuchians" from western central Pangea (now the western portion of North America) have been instrumental in helping untangle the relationships of "rauisuchians" particularly and pseudosuchians in general. Remains of "rauisuchians" occur through the Chinle Formation and Dockum Group (*Long & Murry, 1995*) and now it is clear that nearly all of those taxa or unnamed forms can be sorted into two major groups, the Poposauroidea (*Poposaurus gracilis* and numerous shuvosaurids) and the rauisuchids (*Postosuchus kirkpatricki* and similar forms such as *Viveron haydeni*). To date, these two groups represent highly derived forms within Pseudosuchia and western North America is clearly lacking early diverging paracrocodylomorphs (e.g., *Mandasuchus tanyauchen* from Tanzania), early diverging loricatans (South American or Africa forms like *Prestosuchus chiniquensis*), or more "middle" loricatan forms like *Batrachotomus kupferzellensis* (from Germany). Out of all of the forms from current-day western North America, only one possible taxon fits into this gap *Heptasuchus clarki*, was the first formally recognized "rauisuchian" in North America, but was only briefly described when named (*Dawley, Zawiske & Cosgriff, 1979*). Moreover, *Heptasuchus clarki* occurs in Triassic sediments of central Wyoming, a place that few vertebrates of this age have been

found. Since its naming, *Long & Murry (1995)* reevaluated parts of its anatomy and considered the taxon as a possible synonym of *Poposaurus*, whereas it has been mentioned as a "rauisuchian", but not formally described or placed into a phylogenetic context.

In this article, we fully detail the osteology of *Heptasuchus clarki* by describing the holotype skull and associated postcranial material from the type locality bone bed, provide details on a revised geologic setting and age for the taxon and evaluate its evolutionary affinities with other pseudosuchians.

## METHODS

The holotype and original material collected during the original excavation were collected under with Bureau of Land Management permission. The TMM *Heptasuchus clarki* material was collected under a Bureau of Land Management permit (BLM Survey and Limited Surface Collection Permit PA09-WY-177) facilitated by Dale Hansen and additional help from Brent Breithaupt.

### Geological setting: locality, regional, age and associated assemblage

The *Heptasuchus clarki* type locality (= Clark Locality of *Dawley, Zawiske & Cosgriff, 1979*) occurs within a sequence of red beds near the Red Wall Valley on the southeastern flank of the Big Horn Mountains in central Wyoming (Natrona County) within the Chugwater Group (Fig. 1). The *Heptasuchus clarki* bonebed occurs in a sequence of highly calcareous intraformational conglomerates, thin ripple marked highly bioturbated sandstone beds, silty micrites and reddish brown to dusky red and intercalated green mudstones. All in situ material of *Heptasuchus clarki* (e.g., partial skull, some postcrania) (see below) was derived from 2 to 30 cm thick red mudstone/weathered red regolith (Fig. 1), which is exposed across the bonebed. All cranial elements were found in situ and disarticulated, but closely associated in a one-half square meter (Fig. 2) area in this red mudstone (Fig. 1). Nearly all of the surface collected specimens of *Heptasuchus clarki* and the associated assemblage were collected from the weathered red regolith. With the exception of a lungfish tooth (UW 11567) and a small centrum, no other bones were found below the red mudstone in the underling green mudstone, thin limestone, or conglomerate.

The depositional setting at the locality is inferred to have been a vegetated distal floodplain environment, periodically experiencing sheet floods and the development of ephemeral ponds and lakes. The sheet floods generated the intraformational conglomerates with calcareous nodules and mudstone clasts scoured from soils on the flood plain sediments. The limestone microconglomerates at the *Heptasuchus clarki* site indicate high-energy flood events and the silty micrites suggest post-flood deposition in lakes and ponded, abandoned channels.

The inclusion of the *Heptasuchus clarki* bonebed into a formal stratigraphic unit in the Chugwater Group on the southeastern flank of the Big Horn Mountains has been challenging and debated (*Dawley, Zawiske & Cosgriff, 1979*; *Lucas, Heckert & Hotton, 2002*). These debates are the result of a number of factors including the lack of continuous outcrops in the area, the unique sedimentology of the unit that the *Heptasuchus clarki*

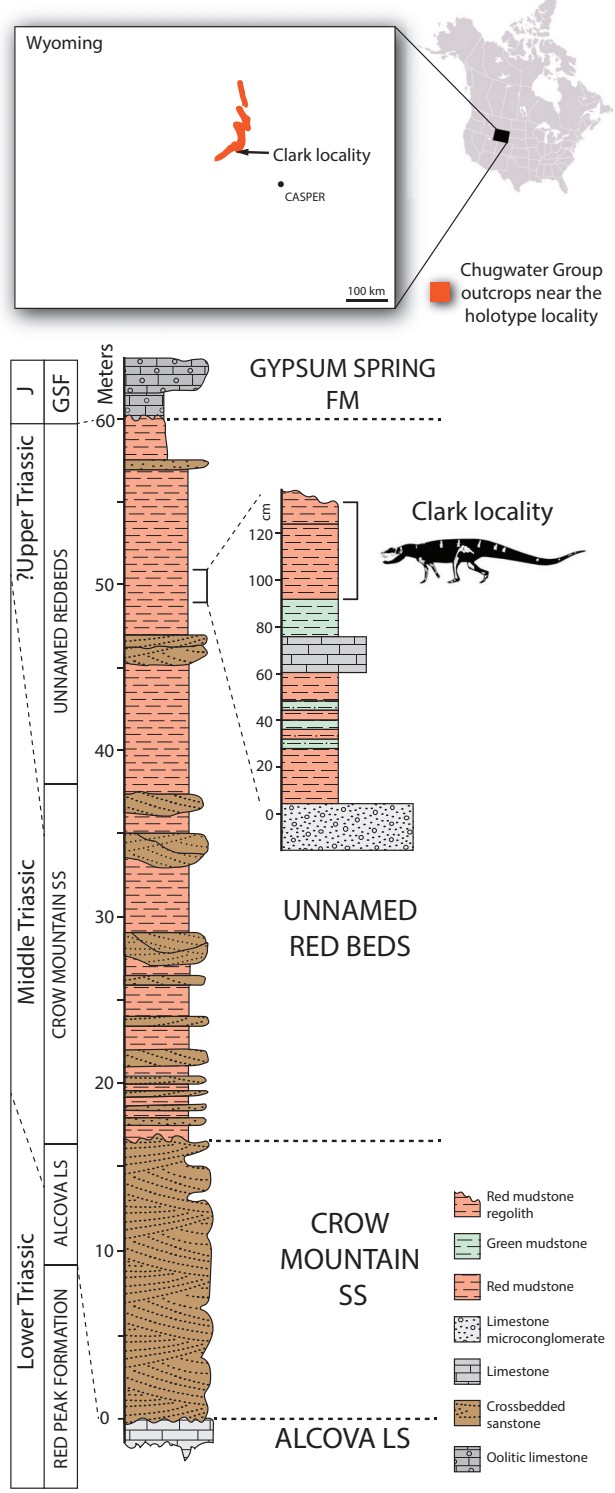

**Figure 1 Distribution of Chugwater Group strata in Wyoming near the location of the type locality of *Heptasuchus clarki*.** Stratigraphic section at the type locality of *Heptasuchus clarki* in the upper portion of the unnamed red beds of the upper portion of the Chugwater Group, Big Horn Mountains and a detailed stratigraphic section through the bonebed. Abbreviations: cm, centimeters; GSF, Gypsum Springs Formation; J, Jurassic; LS, limestone; SS, sandstone.

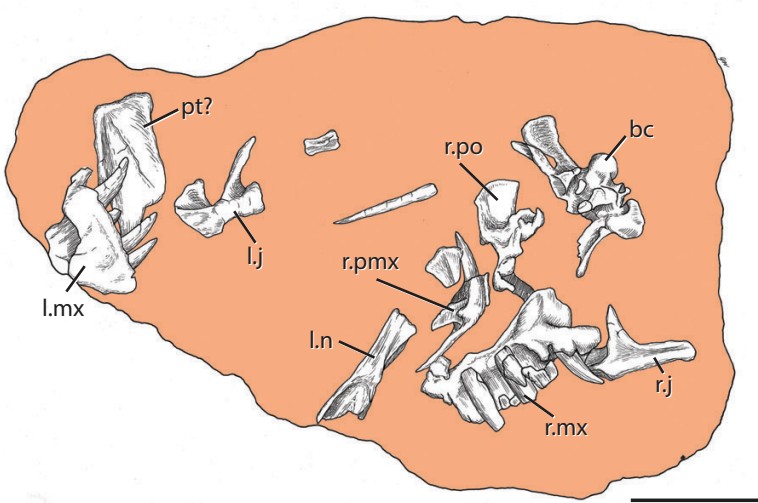

**Figure 2 The holotype skull of *Heptasuchus clarki* (UW 11562) as found in the field.** Image credit: Robert M Dawley. Abbreviations: bc, braincase; j, jugal; l., left; mx, maxilla; n, nasal; pmx, premaxilla; po, postorbital; r., right; pt?, ?pterygoid. Scale = 20 cm.

bonebed lies in, the lack of clear lithostratigraphic signatures of other Triassic formations across Wyoming and the lack of unambiguous, and useful fossils for biostratigraphic correlation. It is clear that the *Heptasuchus clarki* type locality lies well above the Red Peak Formation and the Alcova Limestone given that both crop out locally within a kilometer and can be easily mapped. It is also clear that *Heptasuchus clarki* bonebed lies about 50 m from the top of the Alcova Limestone and ~10 m below the Jurassic-aged Gypsum Springs Formation (Fig. 1) which crops out on a nearby butte (~30 m away).

The strata between the Alcova Limestone and the Gypsum Springs Formation have been assigned to a number of stratigraphic units. The Crow Mountain Sandstone lies directly on the Alcova Limestone and consists of sandstones with current crossbedding (*Cavaroc & Flores, 1991*). The fluvial and lacustrine sediments stratigraphically above the Crow Mountain Sandstone, but below the Gypsum Springs Formation, have been assigned to the Popo Agie Formation based on the stratigraphic position and general lithology (*Picard, 1978*) or by fossil vertebrates from this area (*Dawley, Zawiske & Cosgriff, 1979*; *Lucas, Heckert & Hotton, 2002*) whereas geologists working in the same area assigned these strata to the "unnamed red beds" and hypothesize that the Popo Agie Formation in this region was removed by Jurassic erosion and is not present in the area (*Cavaroc & Flores, 1991*; *Irmen & Vondra, 2000*). The sedimentology and sequence of these strata in question are demonstrably different from that of the Popo Agie Formation further west. *High & Picard (1969)* and *Cavaroc & Flores (1991)* interpreted the lenticular and sheet sandstones in the lower portion of the unnamed red beds as channel and splay deposits of a westward prograding fluvial deltaic plain, comparable to equivalent facies of the Jelm Formation (*Picard, 1978*), specifically the Sips Creek Member of the Jelm Formation of south-central Wyoming (*Pipiringos & O'Sullivan, 1978*; *Blakey, Peterson & Kocurek, 1988*; *Cavaroc & Flores, 1991*). *Cavaroc & Flores (1991)* considered the calcareous sandstones,

silty micrites and red mudstones of the upper portion of the unnamed red beds to be lake deposits that formed in passive areas of a well-integrated alluvial plain. This juxtaposition of fluvial deltaic in the lower portion and the fossiliferous fluvial—lacustrine facies also characterizes the relationship between the Jelm Formation and vertebrate-bearing lower portion of the Popo Agie Formation in the Wind River Range (*High & Picard, 1969*, *Picard, 1978*). The *Heptasuchus clarki* bonebed lies in the fluvial—lacustrine facies in the upper 10 m of the unnamed red beds and there appears to be a clear color transition located just stratigraphically below the locality. Whether this upper part of the unnamed red beds is equivalent to the Popo Agie Formation or part of the Jelm Formation is not clear.

## Age

The age of the *Heptasuchus clarki* bonebed within the unnamed red beds is poorly constrained because of the lack of both unambiguous correlations and biostratigraphically informative fossils. No direct dating methods have been used in the area, but there is a lower bound and upper bound. The Alcova Limestone from the local area was dated as Spathian or earliest Anisian (Aegean) age as suggested by the position of the $^{87}Sr/^{86}Sr$ data on the global marine $^{87}Sr/^{86}Sr$ curve (*Lovelace & Doebbert, 2015*). The sequence is capped by the Gypsum Springs Formation and this has been assigned a Jurassic age (*High & Picard, 1965*; *Pipiringos & O'Sullivan, 1978*). Thus, the Crow Mountain Sandstone and the unnamed red beds are constrained to Middle-Upper Triassic and this has been suggested by many (*High & Picard, 1969*; *Picard, 1978*; *Pipiringos & O'Sullivan, 1978*; *Blakey, Peterson & Kocurek, 1988*; *Cavaroc & Flores, 1991*).

Further constraints on the age of the unnamed red beds was based on lithostratigraphic correlation to units with biostratigraphically informative vertebrates. Historically, this region was correlated with the Upper Triassic Popo Agie Formation from the Wind River Range and this formation has a rich vertebrate record comprised of phytosaurs (*Lees, 1907*; *Mehl, 1915*; *Lucas, 1994*; *Lucas, Heckert & Rinehart, 2007*), metoposaurids (*Branson & Mehl, 1929*), dicynodonts (*Williston, 1904*), and a paracrocodylomorph (*Mehl, 1915*). The presence of metoposaurids and *Parasuchus* has been taken to indicate an early Late Carnian age (*Paleorhinus* Biochron of *Lucas (1998)*; *Lucas, Heckert & Rinehart, 2007*); however, the general validity of such biochrons is currently a contentious issue (*Rayfield, Barrett & Milner, 2009*). Regardless, no clear Popo Agie Formation taxa have been found at the *Heptasuchus clarki* bonebed; no phytosaur teeth or osteoderms and large temnospondyl dermal fragments that are common throughout the Popo Agie Formation were found directly at the locality. Metoposaurid dermal bone fragments and phytosaur teeth (UW 11571) have been found in the area (~5 km) of the *Heptasuchus clarki* bonebed but it is not clear if these occur in the same stratigraphic unit. Furthermore, a *Hyperodapedon* rhynchosaur was found to the north of the *Heptasuchus clarki* bonebed (*Lucas, Heckert & Hotton, 2002*) and the presence of this genus of rhynchosaur was used to argue for an Upper Triassic age for the strata in this area (including the *Heptasuchus clarki* bonebed). However, the correlation of the *Hyperodapedon* locality and

the *Heptasuchus clarki* bonebed is not clear and no diagnostic rhynchosaur remains have been found at the *Heptasuchus clarki* bonebed.

Using what little age constraints are available, the age of the *Heptasuchus clarki* bonebed could range from Middle to Late Triassic. Our best hypothesis concerning the age is that the upper portion of the unnamed red beds at the *Heptasuchus clarki* type locality is equivalent to, or just older than, that of the early Late Triassic Popo Agie Formation assemblage from western Wyoming.

### Associated assemblage

The *Heptasuchus clarki* bonebed has produced the remains of at least four individuals of *Heptasuchus clarki* (see below) as well as bones of much smaller vertebrates; these specimens are represented in collections at UW (e.g., UW 11568-115670), TMM, USMN, and NMMNH. Of the larger vertebrates, we hypothesize that all of the material pertains to *Heptasuchus clarki*, although none of the postcrania is part of the holotype. The criticism that the material represents a mix of a 'rauisuchian' and generically indeterminate phytosaur (*Wroblewski, 1997*) is not supported here given that (1) we have not seen clear evidence that there is more than one "rauisuchian" based on comparisons with *Batrachotomus kupferzellensis* and (2) we have not positively identified any phytosaur crania, teeth, or postcranial material. Of the smaller vertebrates, vertebrae, limb bones, small teeth, and other fragments were abundant on the surface, but nearly all of these elements are broken (e.g., vertebral centra halves, limb bone end). A single lungfish tooth was found at the locality (UW 11567). The identification of this material is ongoing and will be the subject of another publication.

## SYSTEMATIC PALEONTOLOGY

ARCHOSAURIA *Cope, 1869* sensu *Gauthier, 1986*
SUCHIA *Krebs, 1976* sensu *Sereno, McAllister & Brusatte, 2005*
*Heptasuchus clarki Dawley, Zawiske & Cosgriff, 1979*
"*Heptasuchus*"; *Benton, 1986*: 298
"*Heptasuchus clarki*"; *Bonaparte, 1984*: 213
"*Heptasuchus clarki*"; *Parrish, 1993*: 301
"*Heptasuchus clarki*"; *Juul, 1994*: 10
"*Heptasuchus clarkei*"; *Long & Murry, 1995*: 154
"*Heptasuchus clarki*"; *Lucas, 1998*: 364
"*Heptasuchus clarki*"; *Alcober, 2000*: 313
"*Heptasuchus clarki*"; *Gower, 2000*: 451
"*Heptasuchus clarki*"; *Lucas, Heckert & Hotton, 2002*: 150
"*Heptasuchus*"; *Sulej, 2005*: 85
"*Heptasuchus*"; *Lucas, Heckert & Rinehart, 2007*: 222
"*Heptasuchus*"; *Peyer et al., 2008*: 363
"*Heptasuchus*"; *Brusatte et al., 2010*: 10
"*Heptasuchus clarki*"; *De França et al., 2013*: 473
"*Heptasuchus clarki*" *Nesbitt et al., 2013*: 246

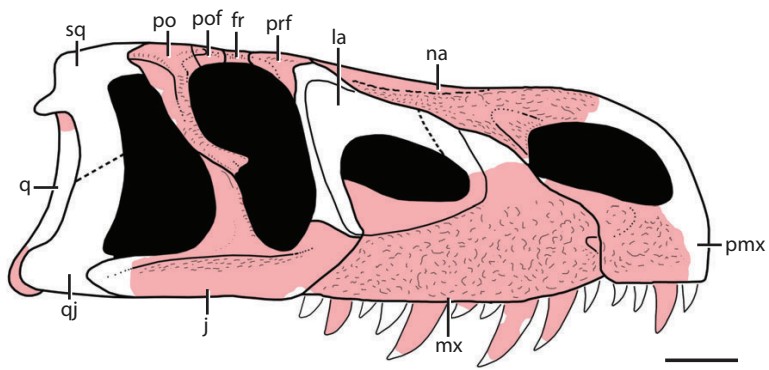

**Figure 3 Reconstruction of the skull of *Heptasuchus clarki* in right lateral view illustrating the material recovered (light red) from the type locality.** Skull reconstruction based on *Batrachotomus kupferzellensis* from *Gower (1999)*. Abbreviations: fr, frontal; j, jugal, la, lacrimal; mx, maxilla, na, nasal; pmx, premaxilla; po, postorbital; pof, postfrontal; prf, prefrontal; q, quadrate; qj, quadratojugal; sq, squamosal. Scale = 5 cm.

**Holotype:** UW 11562, partial skull (Figs. 2–7): right premaxilla (UW 11562-A); right maxilla (UW 11562-B); left maxilla (UW 11562-C); right jugal (UW 11562-D); left jugal (UW 11562-E); right nasal (UW 11562F); right postfrontal, postorbital, partial frontal, and prefrontal (UW 11562-G); occiput and braincase (UW 11562-H); left palatine (UW 11562-K); pterygoid? (UW 11562-L); pterygoid fragment (UW 11562-M); fragment of hyoid? (UW 11562-N); unidentified skull fragments (UW 11562-O through -R); loose teeth (UW 11562-AA through -AI). Here, the holotype is restricted to the cranial elements found in situ in quad A-3 (Fig. 2). No skull element is duplicated and the relative similar sizes of the elements suggest that the remains are from a single individual.

**Referred material:** quadrate head (UW 11563-AD); ventral condyles of left quadrate (UW 11563-AF, UW 11563-H); anterior cervical vertebra (UW 11562-T) ; posterior cervical centrum (UW 11564-A); posterior trunk vertebra (TMM 45902-2); neural spine of a cervical-trunk vertebra (UW 11562-CX); presacral neural spine (UW 11562-V); presacral neural spine (UW 11562-CT); anterior caudal vertebra (UW 11562-U); distal caudal vertebra (UW 11562-BW; UW 11563-A-C); osteoderm (TMM 45902-1); right partial scapula (UW 11565-E); right partial scapula (UW 11566-B); partial left coracoid (UW 11566); proximal portion of left humerus (UW 11565-A); left humerus (UW 11563-U); proximal portion of the radius (UW 11562-DM); distal portion of the radius (UW 11562-DI; UW 11562-DF); right ulna (UW 11562-W); left ulna (UW 11562-X); distal ends of ulnae (UW 11563-V; UW 11565-C); left pubis (UW 11562-Y); ilium fragment (UW 11563-Y); pubic peduncle of the right ilium (UW 11563-Z); left pubis (UW 11562-Y); proximal portion of the right ischium (UW 11564-B); proximal portion of a right femur (UW 11563-B); distal portion of the right femur (UW 11563-A); left tibia (UW 11562-Z); proximal portion of a right fibula (UW 11566-S); distal portion of the right fibula (UW 11566-R); proximal end of metatarsals (UW 11562-DH, UW 11562-DHU, UW 11562-DR); ungual (UW 11562-DT).

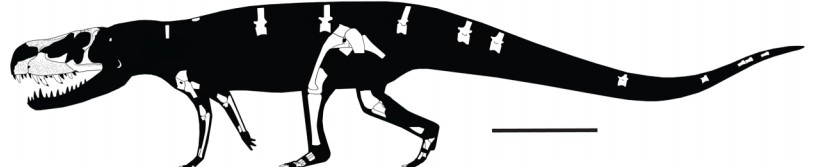

**Figure 4 Reconstruction of the skeleton of *Heptasuchus clarki* in lateral view illustrating the material recovered from the type locality.** Skeleton reconstruction based on *Postosuchus kirkpatricki* (*Nesbitt et al., 2013*) and skull reconstruction based on Fig. 3. Scale = 50 cm.

**Type Locality:** Clark locality; section 2l, TAON, RSQW, E Natrona County, Red Wall Valley, southern Big Horn Mountains, Wyoming, U.S.A.

**Stratigraphic Occurrence:** unnamed red beds of the Chugwater Group. Age = ?Middle Triassic to Upper Triassic (see above for details).

**Differential diagnosis:** *Heptasuchus clarki* differs from all other suchians except for *Batrachotomus kupferzellensis* in possessing the following combination of character states: exit for cranial nerve V within prootic (shared with *Postosuchus*); a depression on the anterolateral surface on the ventral end of the postorbital (character 425 - state 1); a deep depression on the posterodorsal portion of the lateral surface of the ventral process of the postorbital (427-1); a distinct fossa with a rim present on the nasal at the posterodorsal corner of the naris (430-1); the anteroventral corner of the maxilla extensively laterally overlaps the posteroventral corner of the premaxilla (431-1); and an anteroposteriorly trending ridge on the lateral side of the jugal that is asymmetrical dorsoventrally where the dorsal portion is more laterally expanded than the ventral portion (433-1). Furthermore, *Heptasuchus clarki* and *Batrachotomus kupferzellensis* share the following two homoplastic characters within Archosauria: Concave anterodorsal margin at the base of the dorsal process of the maxilla (25-1); and dorsolateral margin of the anterior portion of the nasal with distinct anteroposterior ridge on the lateral edge (35-1: Rauisuchidae synapomorphy also).

 *Heptasuchus clarki* differs from *Batrachotomus kupferzellensis* in that *Heptasuchus clarki* lacks a division in the fossa between the basitubera and basipterygoid processes (=median pharyngeal recess) of the parabasisphenoid, the presence of large and posteriorly pointed processes on the posterior portion of the basipterygoid processes\*; paroccipital processes more broadly expanded distally; no kink in the ventral process of the postorbital (note, not all *Batrachotomus kupferzellensis* specimens have the kink e.g., SMNS 52970); anterior portion of the maxilla is less expanded and has a smaller foramen between maxilla and the premaxilla; palatal process of premaxilla is more expanded medially; palatal process of the maxilla continuous with anterior edge of maxilla (the palatal process is hidden under a flange of bone laterally in *Batrachotomus kupferzellensis*); and the anterolateral corner of postfrontal of *Heptasuchus clarki* is blunt and squared off in dorsal view\*. Asterisks denote autapomorphies of *Heptasuchus clarki*.

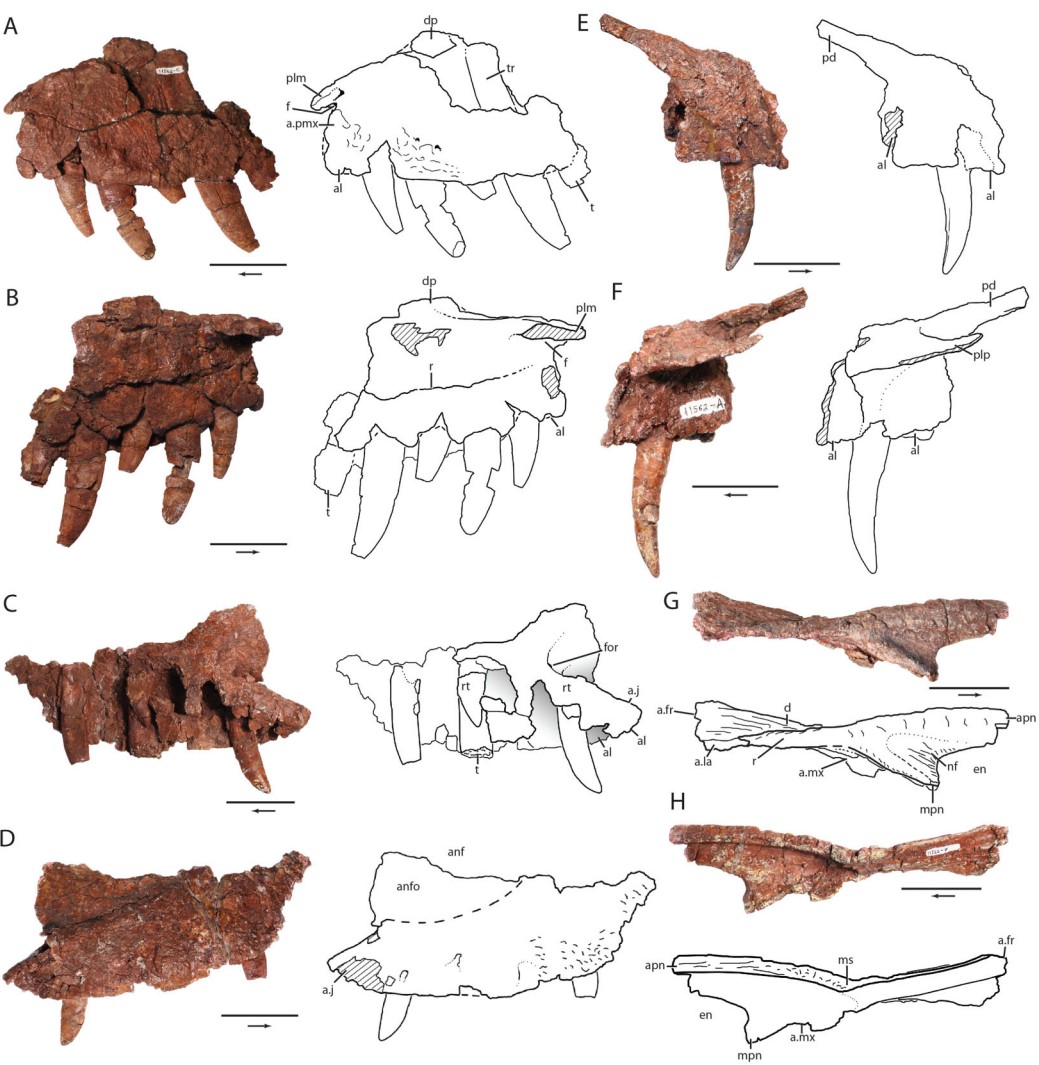

**Figure 5 Skull elements of *Heptasuchus clarki* (UW 11562).** Left maxilla (UW 11562-C) in lateral (A) and medial (B) views; right maxilla (UW 11562-B) in medial (C) and lateral (D) views; right premaxilla (UW 11562-A) in lateral (E) and medial (F) views; right nasal (UW 11562-F) in lateral (G) and medial (H) views. Abbreviations: a., articulates with; al, alveolus; anf, antorbital fenestra; anfo, antorbital fossa; apn, anterior process of nasal; d, depression; dp, dorsal process; en, external naris; f, fossa; for, foramen; fr, frontal; j, jugal; la, lacrimal; ms, midline suture; mx, maxilla; mpn, maxillary process of nasal; nf, narial fossa; pd, posterodorsal process; plp, palatal process of the premaxilla; plm, palatal process of the maxilla; pmx, premaxilla; r, ridge; rt, replacement tooth; t, tooth; tr, tooth root. Broken surfaces indicated in hash marks. Arrows indicate anterior direction. Scales = 5 cm.

**Ontogenetic status**: The ontogenetic stages of the specimens of *Heptasuchus clarki* are difficult to assess given the holotype contains only skull elements and the postcrania of the taxon has poor association with cranial or other postcranial remains. An ontogenetic age assessment based on the skull (e.g., fusion events) is not reliable in archosaurs (*Bailleul et al., 2016*). With the exception of a complete tibia and nearly complete ulna, no other limb bones, such as the femoral fragments have a midshaft that could be used for histological analysis. Fragments of limb bones are available, even so, identification of the
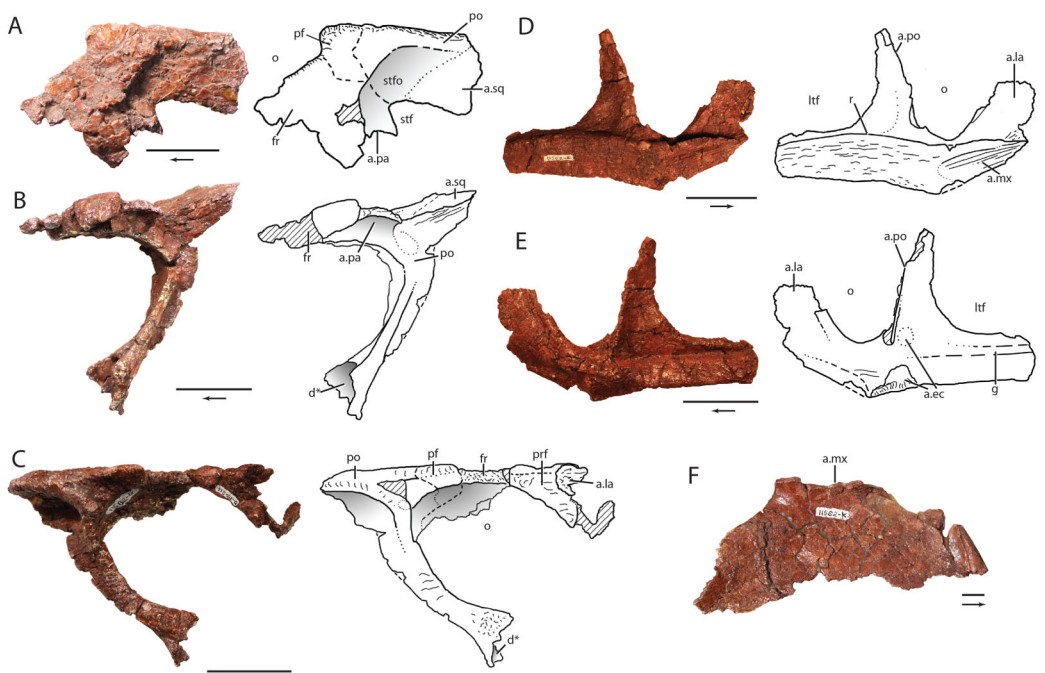

**Figure 6 Skull elements of *Heptasuchus clarki* (UW 11562).** Right postorbital, postfrontal, and frontal (UW 11562-G) in dorsal (A), medial (B) and, with the reattached prefrontal in lateral (C) views; right jugal (UW 11562-D) in lateral (D) and medial (E) views; left palatine (UW 11562-K) in dorsal (F) view. Abbreviations: a., articulates with; d, depression; ec, ectopterygoid; f, fossa; fr, frontal; g, groove; la, lacrimal; ltf, lower temporal fenestra; mx, maxilla; o, orbit; pa, parietal; pf, postfrontal; po, postorbital; prf; prefrontal; sqm squamosal; r, ridge; stf; supratemporal fenestra; stfo, surpratemporal fossa. Broken surfaces indicated in hash marks. Arrows indicate anterior direction. Scales = 5 cm.

element based on a limb shaft is difficult and the orientation of the fragments and overall size of the limb would be difficult to assess for comparative purposes. Of the few vertebrae recovered, all neurocentral sutures appear to be fully closed (*Brochu, 1996*; *Irmis, 2007*). This is clear in the partial cervicals, trunk and anterior caudal vertebrae. Based on this cursory assessment, the specimens of *Heptasuchus clarki* are not young individuals, but their ontogenetic stage is largely unconstrained with the available evidence.

**Notes**: The original holotype of *Heptasuchus clarki* (*Dawley, Zawiske & Cosgriff, 1979*) was amended by *Zawiskie & Dawley (2003)*, who restricted it to the in situ cranial material collected in 1977 in quads A-1 and A-2 of the excavation grid at the Clark locality (see grid in *Dawley, Zawiske & Cosgriff, 1979*) following the criticism that the taxon may represent a chimera (*Wroblewski, 1997*). Much of the bonebed was weathered and many bone fragments littered the ground and these specimens were collected in 1977–1979 and in 2009–2010. The association of the postcranial elements is not known but are assigned to *Heptasuchus clarki* based on similarity among elements and similarity to the almost completely known anatomy of *Batrachotomus kupferzellensis* (*Gower, 1999*; *Gower & Schoch, 2009*); we are assuming that all of the archosaur material that is similar in comparative size emanates from a single taxon of loricatan. Therefore, we only refer

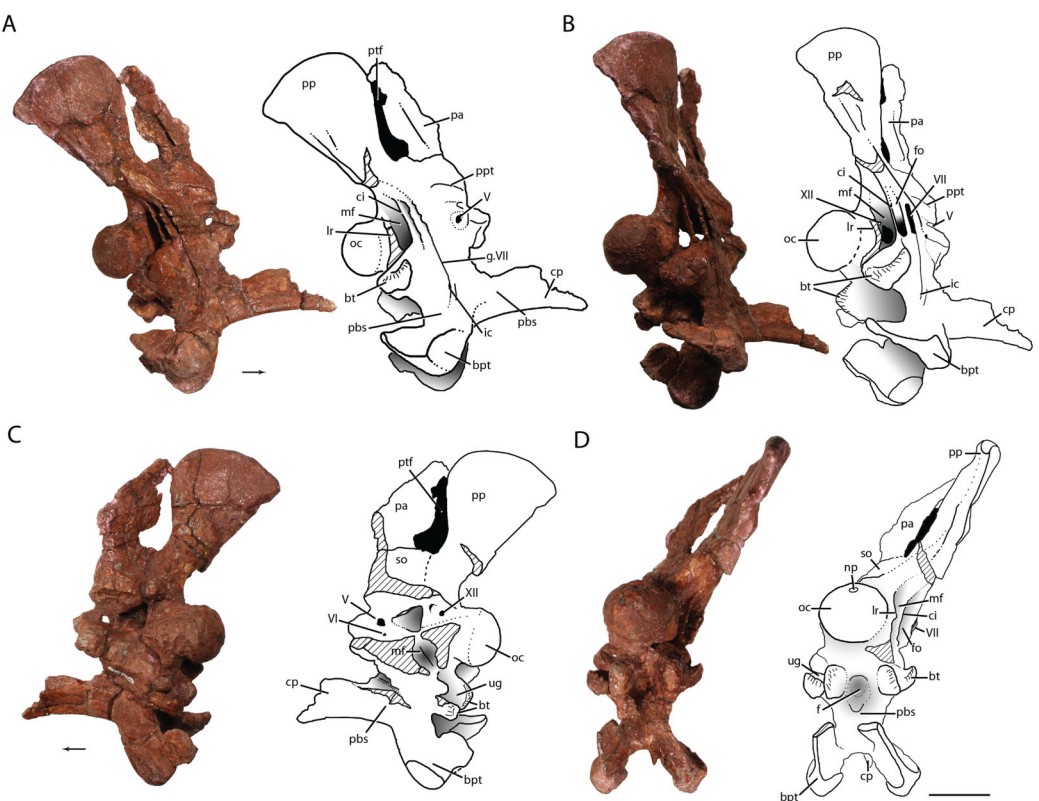

**Figure 7 The braincase of *Heptasuchus clarki* (UW 11562-H) in right lateral (A), posterolateral (B), medial (C) and posterior (D) views.** Abbreviations: bt, basitubera; bpt, basipterygoid process; ci, crista interfenestralis; cp, cultriform process; f, fossa; fo, fenestra ovalis; g., groove for; ic, entrance of the internal carotid; lr, lateral ridge; mf, metotic foramen; np, notochoral pit; oc, occipital condyle; pa, parietal; pbs, parabasisphenoid; pp, paroccipital process of the otoccipital; ppt; ridge possibly for attachment of protractor pterygoidei; ptf, posttemporal fenestra; so, supraoccipital; ug, unossified gap; V, exit of cranial nerve V (trigeminal); VI, exit of cranial nerve VI (abducens); VII, exit of cranial nerve VII (facial); XII, exit of cranial nerve XII (hypoglossal). Broken surfaces indicated in hash marks. Arrows indicate anterior direction. Scales = 5 cm.

material to the taxon and do not create paratype specimens. The locality has a minimum number of three individuals (MNI = 3) of similar size, as deduced from the number of right distal ends of the ulna.

**Comparative Morphological Description**
**General skull:** Most of the skull of the holotype specimen (UW 11562-A through -S) was recovered as separate, disarticulated bones, except for the postorbital-postfrontal-frontal prefrontal section and the braincase-parietal. The total complement of bones is by no means complete and several elements (lacrimal, squamosal, quadratojugal, and quadrate) are not represented on either the right or left side. However, sufficient material is preserved to provide a reconstruction of most areas of the skull (Fig. 3) and skeleton (Fig. 4). Only the quadrate region is totally unknown, and the palate is represented only by a single fragment. We estimate the skull to be about 56 cm long.

The following describes the general aspect of the skull and details of each element are included below. The skull is long and narrow with the preorbital (tooth-bearing) length about two-thirds that of the total length. In lateral view (Fig. 3), the lower margin of the skull forms, roughly, an obtuse angle whose apex points ventrally and is located at the level of the sixth maxillary tooth. There are three premaxillary and nine maxillary teeth preserved. A small subnarial fenestra is present between the premaxilla and the maxilla (see more details below), but this area is damaged. Posteriorly, a moderately large antorbital fenestra lies in a recessed antorbital fossa. The orbit is 'keyhole shaped,' and this configuration reflects the expansion of the lower part of the enlarged infraorbital fenestra. In the area of the nasal, the lateral borders of the skull roof form a pair of elevated ridges, which flank a shallow depression in the center of the dorsal surface of the skull roof. The supratemporal fenestra is small, triangular, and surrounded by a supratemporal fossa.

**Premaxilla:** The premaxilla is only known from the right side (UW 11562-A; Figs. 5E and 5F) and lacks the anterior portion of the first preserved alveolus, the posterior end of the third alveolus, and the entirety of the anterodorsal (=narial) process. *Heptasuchus clarki* was originally described as having three premaxillary teeth, but the tooth-bearing margin is incomplete. At least three premaxillary teeth are present, but the exact number of premaxillary teeth is unknown. The body of the premaxilla is rounded laterally and does not preserve a distinct narial fossa anteroventral to the external naris, a distinct feature of the premaxilla of *Batrachotomus kupferzellensis*. No foramina are apparent on the premaxilla, but this is possibly the result of a highly fractured and partially weathered surface.

Two prominent processes are preserved, a palatal and a posterodorsal (=maxillary) processes. The posterodorsal process is straight, slender, and projects 30° posterodorsally. The posterodorsal edge of the process forms a concave margin that frames part of the posterior margin of the external naris. The relative length of the process compared to the length of the premaxillary body is similar to that of *Postosuchus kirkpatricki* (TTUP 9000) and *Rauisuchus tiradentes* (BSPG AS XXV-60-121), longer than that of *Batrachotomus kupferzellensis* (Gower, 1999), and is much shorter than the longer, more robust, and arched subnarial processes present in *Saurosuchus galilei* (PVSJ 32) and *Luperosuchus fractus* (PULR 04; Nesbitt & Desojo, 2017). A small foramen is located in the body of the premaxilla ventral to the base of the posterodorsal process. The base of the posterodorsal process is not laterally expanded into a bulge posteroventral of the external naris as in *Rauisuchus tiradentes* (BSPG AS XXV-60-121; Lautenschlager & Rauhut, 2015), *Vivaron haydeni* (Lessner et al., 2016), *Postosuchus kirkpatricki* (Weinbaum, 2011), and *Polonosuchus silesiacus* (Sulej, 2005).

The palatal process is a broad, flat, transversely oriented sheet of bone that originates at the dorsal margin of the tooth row and projects medially to contact its antimere. Ventrally, the palatal process forms the base of a ventrally opening fossa. The process forms the anterior edge of the anterior portion of the palate, as in *Saurosuchus galilei* (Alcober, 2000). The posterior edge of the process articulates with the vomer.

**Maxilla:** The posterior two-thirds of the right maxilla (UW 11562-B; Figs. 5C and 5D) and the anterior half of the tooth-bearing portion of the left maxilla (UW 11562-C; Figs. 5A and 5B) are present in the holotype of *Heptasuchus clarki*. Only the base of the dorsal (=ascending) process is preserved. The left maxilla preserves the first six alveoli and the preserved portion of the right maxilla preserves eight alveoli. As reconstructed (Fig. 3; *Dawley, Zawiske & Cosgriff, 1979*), a complete maxilla would have a minimum of ten teeth, as determined by overlap of the two preserved maxillae; the anteriormost alveolus from the left maxilla fragment is considered to be equivalent to the anteriormost alveolus of the right maxillary fragment. As reconstructed, the maxilla is a massive, rectangular bone with a deep body similar to that of *Fasolasuchus tenax* (PVL 3851), *Batrachotomus kupferzellensis* (SMNS 80260) and *Saurosuchus galilei* (PVSJ 32).

The anterior portion of the maxilla is well preserved. The lateral surface is rather flat and not laterally expanded. The anterior margin of the maxilla is convex. A small notch is present where the anterolateral portion of the maxilla meets its palatal process. This notch is similar to that of *Batrachotomus kupferzellensis* (SMNS 52970), *Saurosuchus galilei* (PVSJ 32), *Fasolasuchus tenax* (PVL 3851), and *Postosuchus kirkpatricki* (TTUP 9000). In these taxa, a foramen is formed between the articulation of the premaxilla and maxilla when in articulation; this morphology was discussed at length by *Gower (2000)* and *Nesbitt (2011)*. *Heptasuchus clarki* was originally reported (*Dawley, Zawiske & Cosgriff, 1979*) to have an elongated fenestra between the maxilla and premaxilla, similar to what was reported in *Saurosuchus galilei* (PVL 2062; *Reig, 1959*) and *Luperosuchus fractus* (PULR 04; *Romer, 1971*). However, it appears that these elongate fenestrae are the result of disarticulation or deformation (see *Nesbitt, 2011*; *Nesbitt & Desojo, 2017*). Therefore, the elongated fenestra reconstructed in *Heptasuchus clarki* (Fig. 2 of *Dawley, Zawiske & Cosgriff (1979)*) is likely not present. An anteriorly opening foramen is present within the notch between the lateral side of the maxilla and the palatal process which is also found in *Postosuchus kirkpatricki* (*Weinbaum, 2011*). Another, smaller anteriorly opening foramen is located just posterodorsal to the foramen in the notch. The transition between the lateral side of the maxilla and the palatal process is continuous as in *Postosuchus kirkpatricki* (TTUP 9000) and *Fasolasuchus tenax* (PVL 3851), a condition in contrast to *Batrachotomus kupferzellensis* (SMNS 52970) where there is a distinct step. There is no clear facet on the anterodorsal surface of the maxilla for the posterodorsal process of the premaxilla. Here, the surface is incompletely preserved but appears to be concave in lateral view between the palatal process and the base of the dorsal process, as in *Batrachotomus kupferzellensis* (SMNS 52970). It is unknown if the mediolaterally compressed ridge of bone that forms the anterodorsal margin of the maxilla contributed to the border of the external naris as it does in *Batrachotomus kupferzellensis* (*Gower, 1999*). The base of the dorsal process is oval in cross-section, similar to what is present in *Batrachotomus kupferzellensis* (SMNS 52970) and *Arizonasaurus babbitti* (MSM 4590) rather than the anteroposteriorly elongated cross-sections of taxa such as *Postosuchus kirkpatricki* (TTUP 9000).

The entire lateral side of the maxilla ventral to the antorbital fossa is covered in small ridges and shallow grooves much like that in the holotype of *Saurosuchus galilei*

(PVL 2062). A slight bank marks the division of the antorbital fossa from the main body of the maxilla as in *Fasolasuchus tenax* (PVL 3851), *Batrachotomus kupferzellensis* (SMNS 52970), and *Saurosuchus galilei* (PVSJ 32) and not separated by a distinct step as in *Polonosuchus silesiacus* (ZPAL Ab III/563) and *Postosuchus kirkpatricki* (TTUP 9000). The depth of the antorbital fossa deepens posteriorly in *Heptasuchus clarki* as well as *Fasolasuchus tenax* (PVL 3851), *Batrachotomus kupferzellensis* (SMNS 52970), *Saurosuchus galilei* (PVSJ 32), a specimen referred to *Prestosuchus* (UFRGS-PV 156 T), and in the crocodylomorph *Dromicosuchus grallator* (NCSM 13733). The posterior portion of the maxilla expands dorsally as in *Turfanosuchus dabanensis* (IVPP V33237) and gracilisuchids, unlike most loricatans. The bone that forms the antorbital fossa is thin posteriorly as in *Postosuchus kirkpatricki* (TTUP 9000), *Fasolasuchus tenax* (PVL 3851), *Batrachotomus kupferzellensis* (SMNS 52970) and other archosaurs (e.g., *Xilousuchus sapingensis*, IVPP V6026). The tooth bearing ventral margin is convex for the length of the element as in *Batrachotomus kupferzellensis* (SMNS 52970).

The first alveolus is the smallest in the maxilla as is typical for most taxa classically grouped as "rauisuchians" (*Brusatte et al., 2009*). The alveoli increase in size posteriorly to the fourth and fifth alveolus then gradually decrease in size posteriorly based on our reconstructed maxilla from the two pieces. All of the alveoli are ovate in ventral view.

In medial view, a step separates the medial surface of the maxilla from the interdental plates. The step in horizontally oriented and extends the length of the preserved section of maxilla. Anteriorly, the step is located in the dorsoventral middle of the body of the maxilla as in *Fasolasuchus tenax* (PVL 3851) and *Batrachotomus kupferzellensis* (SMNS 52970) whereas the step is located in the ventral third of the anteromedial surface of the maxilla of *Postosuchus kirkpatricki* (TTUP 9000). The anteriormost portion of the step disappears posterior to the anterior termination of the maxilla in *Heptasuchus clarki*. The palatal process is horizontally oriented at the anterodorsal portion of the maxilla. The process is thin dorsoventrally as in *Fasolasuchus tenax* (PVL 3851) and *Batrachotomus kupferzellensis* (SMNS 52970) whereas the process is dorsoventrally deeper in *Postosuchus kirkpatricki* (TTUP 9000). A distinct fossa on the ventral surface of the palatal process is present in *Heptasuchus clarki* and a similar deep fossa is on the ventral surface of the palatal process is also present in *Polonosuchus silesiacus* (ZPAL Ab/III 563), *Fasolasuchus tenax* (PVL 3851), *Batrachotomus kupferzellensis* (SMNS 52970), and the crocodylomorph *Sphenosuchus acutus* (SAM 3014), but absent in *Saurosuchus* (PVSJ 32) and poposauroids (e.g.,, *Xilousuchus sapingensis*) (see character 426 in the appendix). Along the ventral half of the medial surface of the tooth row, the internal walls of the alveoli are formed by fused interdental plates at least anteriorly. The interdental plates of all *Batrachotomus kupferzellensis* specimens (e.g., SMNS 52970) are unfused and separated as triangular sheets of bone), whereas the interdental plates of *Postosuchus kirkpatricki* (TTUP 9000) and *Teratosaurus suevicus* (NHMUK 38646) are rectangular and meet on their anterior and posterior sides and this contact extends to the ventral extent of the medial side of the the maxilla. A thin line marks the contact of the interdental plates in *Postosuchus kirkpatricki* (*Weinbaum, 2011*; TTUP 9000, 9002) whereas there is no differentiation between the individual plates in *Teratosaurus suevicus* (NHMUK 38646).

The loss of the medial surface on the posterior half of the maxilla has exposed the tips of replacement teeth medial to the roots of the fully erupted teeth. Posteriorly, the maxilla separates into two portions, a ventral portion that houses the alveoli and a mediolaterally thin dorsal portion. The ventral portion tapers posteroventrally and expands more posteriorly than the thin dorsal portion. A posteriorly opening foramen lies at the juncture of the ventral and dorsal portions. Here, a faint facet for the articulation with the jugal can be followed posteriorly on the dorsal surface of the maxilla.

**Nasal:** A nearly complete right nasal (UW 11562-F) is known for *Heptasuchus clarki* (Figs. 5G and 5H); only the anterior portion that meets the anterodorsal (=nasal) process of the premaxilla is missing. The nasal formed the posterodorsal portion of the external nares. The anterior half of the nasal splits into a robust anterior process that would have met the anterodorsal process of the premaxilla, if complete, and a shorter, anteroventrally directed process that lies on the anterodorsal margin of the maxilla. The anterior process bows dorsally to form a "roman nose" similar to that of *Batrachotomus kupferzellensis* (*Gower, 1999*), *Saurosuchus galilei* (PVSJ 32), a skull assigned to *Prestosuchus chiniquensis* (UFRGS T-156), *Luperosuchus fractus* (PULR 04; *Nesbitt & Desojo, 2017*), and *Decuriasuchus quartacolonia* (*De França, Ferigolo & Langer, 2011*). The lateral surface of the anterior process bears a rugose lateral ridge that continues posteriorly to the articular surface with the lacrimal. This ridge is similar to that in *Postosuchus kirkpatricki* (TTUP 9000) and *Batrachotomus kupferzellensis* (*Gower, 1999*). A distinct fossa is present posterodorsal to the external naris at the junction of the anterior process and the anteroventral process. The fossa is well defined and is similar to that of *Batrachotomus kupferzellensis* (*Gower, 1999*) (see character 430) and an isolated nasal fragment (NMMNH 55779) from the Middle Triassic Moenkopi Formation of New Mexico (*Schoch et al., 2010*). The anteroventral process tapers just ventral to the posterior extent of the external naris. The location of the anterior termination of this process is not known and it is not clear if the process met the posterodorsal process of the premaxilla, hence excluding the maxilla from the external naris, as in the case in *Batrachotomus kupferzellensis* (*Gower, 1999*).

The nasal articular surfaces with the maxilla and lacrimal lie at the ventrolateral edge and are oriented almost vertically, indicating a nearly perpendicular contact between these bones and the nasal. It appears that the nasal formed the anterodorsal portion of the antorbital fossa in *Heptasuchus clarki* as in *Batrachotomus kupferzellensis* (*Gower, 1999*) but not in *Postosuchus kirkpatricki* (TTUP 9000) or *Saurosuchus galilei* (PVSJ 32). Dorsally, the surface medial to the lateral ridge is dorsoventrally thin and concave at the midline like that of *Postosuchus kirkpatricki* (TTUP 9000, 9002), *Batrachotomus kupferzellensis* (*Gower, 1999*), the crocodylomorph *Sphenosuchus acutus* (*Walker, 1990*) and *Turfanosuchus dabanensis* (IVPP V3237). This concave depression narrows anteriorly until it disappears just posterior to the division of the anterior portion of the nasal.

The medial surface of the nasal bears a dorsoventrally thick midline suture that thins posteriorly. The suture itself bears a series of complex grooves and ridges. The medial

surface is largely concave anteriorly and flat posteriorly where the nasal is dorsoventrally thin.

**Jugal:** Both the right and left jugals of *Heptasuchus clarki* are represented in the holotype (UW 11562-D and -E, respectively; Figs. 6D–6F). The right jugal is missing the dorsal end of the ascending process and the posterior portion of the posterior process whereas the left element is missing much of the posterior process. The jugal is a triradiate structure, with two dorsal processes contributing to the ventral portions of the anterior and posterior walls of the orbit and a posterior process forming much of the lower margin of the infratemporal fenestra. The anterodorsal process projects forward at approximately 50° anterodorsally along its contact with the maxilla. Elongated groove and ridges mark the articulation with the maxilla and this articulation terminates posteriorly in an acute angle within the body of the jugal. A similar termination within the jugal is present in *Batrachotomus kupferzellensis* (SMNS 52970) as well as *Revueltosaurus callenderi* (PEFO 34561) and aetosaurs (*Nesbitt, 2011*). The anterodorsal process trends mediolaterally in the dorsal direction where it would meet the lacrimal. The articular surfaces with the maxilla and the lacrimal are separated by a distinct anteroposteriorly trending ridge that continues posteriorly as the laterally expanded jugal ridge. Anteriorly, this ridge is sharp, mediolaterally thin, hides parts of the lateral side of jugal in lateral view, and dorsally forms a small shelf. A similar shelf is present in *Batrachotomus kupferzellensis* (SMNS 52970) and definitely absent in *Postosuchus kirkpatricki* (TTUP 9000) and *Saurosuchus galilei* (PVSJ 32). The anterior surface shifts vertically at the anterior edge, and terminates in a sutural surface with the lacrimal. The lacrimal appears to have articulated with the lateral side of the jugal but the details of this articulation are not clear.

The prominent jugal ridge on the lateral side of the body of the jugal of *Heptasuchus clarki* continues for the length of the jugal. The lateral side of the ridge is covered in small anteroposteriorly trending ridges and lacks the long grooves present in *Batrachotomus kupferzellensis* (SMNS 52970). In its anteroposterior center, the lateral ridge is asymmetrical with the dorsal portion more laterally expanded then the ventral portion. This asymmetry is also present *Batrachotomus kupferzellensis* (SMNS 52970) whereas other paracrocodylomorphs (e.g., *Postosuchus kirkpatricki*, *Saurosuchus galilei*) have a dorsoventrally symmetrical lateral ridge. The posterior process is rectangular in cross-section and the ventral edge of the jugal is nearly straight.

The dorsal process of the jugal arcs posterodorsally at its dorsal termination. The lateral side bears a shallow fossa at the base and on the posterior half of the process. A similar fossa is also present in *Batrachotomus kupferzellensis* (SMNS 52970). The anterior edge of the dorsal process is mediolaterally thin and distinctly convex as in *Batrachotomus kupferzellensis* (SMNS 52970) whereas the anterior edge is typically straight in other loricatans (e.g., *Postosuchus kirkpatricki*; TTUP 9000). The anterior bowing of the anterior edge of the dorsal process of *Heptasuchus clarki* suggests that the ventral portion of the orbit is more anteroposteriorly restricted than the dorsal portion of the orbit. Therefore, it is clear that *Heptasuchus clarki* had a 'keyhole shaped' (sensu *Benton & Clark, 1988*) orbit as with non-crocodylomorph loricatans and other large carnivorous archosaurs (e.g., allosaurids, tyrannosaurids). In *Heptasuchus clarki*, the thin anterior margin hides the
articular surface with the postorbital. The concave posterior margin of the process is mediolaterally thin. Overall, the dorsal process is subcircular in cross-section at its base.

Medially, the body of the jugal is convex anteriorly and concave posteriorly. The posterior process bears an anteroposteriorly oriented groove that is also present in the loricatans *Batrachotomus kupferzellensis* (SMNS 52970), *Postosuchus kirkpatricki* (TTUP 9000), some crocodylomorphs (e.g., *Sphenosuchus acutus*, *Walker, 1990*) and in phytosaurs (*Stocker, 2010*; *Stocker & Butler, 2013*). Anteriorly, just ventral to the dorsal process, the groove divides the articular facets for the ectopterygoid. The head of the ectopterygoid likely split into two lateral heads as with *Batrachotomus kupferzellensis* (SMNS 80260), *Postosuchus kirkpatricki* (*Weinbaum, 2011*), and crocodylomorphs (e.g., *Sphenosuchus acutus*, *Walker, 1990*). The dorsal articular surface for the ectopterygoid is round and poorly defined whereas the ventral articulation is well defined and extends to the ventral edge of the jugal. The articular surface with the postorbital lies on the anteromedial edge of the dorsal process and extends ventrally for much of the length of the dorsal process. Therefore, the anterior edge is mediolaterally thick. Anteriorly, the jugal has a shallow fossa on the ventral edge, opposite the articular facets. A small channel is present between the fossa and the ventral articular surface with the ectopterygoid.

**Skull roof elements:** A large fragment of the skull roof (UW 11562-G) comprises the right prefrontal, postfrontal, frontal, and postorbital (Figs. 6A–6C). With the exception of the frontal, the elements are essentially complete, but microfracturing has obscured the sutural contacts between them.

The prefrontal (Fig. 6C) lies on the anterolateral edge of the frontal and forms the anterodorsal corner of the orbit. The lateral margin bears a rugose lateral ridge that could have been continued from the nasal to the lacrimal to the prefrontal as in rauisuchids and *Batrachotomus kupferzellensis*. The posterolateral margin of the prefrontal does not have a clear sutural contact for a supraorbital element or palpebral(s) that are present on the prefrontal in *Saurosuchus galilei* (PVSJ 32) and *Postosuchus kirkpatricki* (TTUP 9000; *Weinbaum, 2011*; *Nesbitt, Turner & Weinbaum, 2013*). A rugose articulation with the lacrimal located on the anterior portion of the prefrontal is inset from the lateral margin and rounded posteriorly. The ventral end of the prefrontal is broken.

The anterior and medial parts of the frontal are incomplete (Fig. 6). The frontal clearly contributes to the lateral margin of the orbit. Here, the lateral orbital margin is rounded and slightly rugose. The preserved portion of the dorsal surface of the frontal is smooth, but much of the surface is poorly preserved and fragmented. The suture between the postfrontal and the frontal is clear on the ventral surface of the elements. Posteriorly, it appears that part of the supratemporal fossa is present on the frontal as in crocodylomorphs, dinosaurs, and *Batrachotomus kupferzellensis* (SMNS 80260) (*Nesbitt, 2011*). In *Postosuchus kirkpatricki* (TTUP 9000), a supratemporal fossa is present anterior to the supratemporal fenestra, but present almost entirely on the postfrontal (*Weinbaum, 2011*; *Nesbitt, 2011*) with only the medial portion of the frontal participating in the fossa. Thus, among non-crocodylomorph loricatans, a fossa on the posterior portion

of the frontal seems to be restricted to *Heptasuchus clarki, Batrachotomus kupferzellensis*, and *Postosuchus kirkpatricki*. The posterior edge of the frontal appears to contribute to the border of the supratemporal fenestra.

The postfrontal lies at the posterodorsal edge of the orbit (Fig. 6). In dorsal view, the anterolateral corner angle is nearly 90° from the anterior orbital margin to the lateral margin. The anterior and the lateral edges of the element are rounded and have small grooves on them. The body of the postfrontal dorsally overhangs the postorbital where the two elements meet. The medial portion tapers posteromedially between the frontal and the postorbital, and apparently is not part of the supratemporal fossa.

The postorbital is nearly completely preserved (Fig. 6). The postorbital has two components, a dorsal portion, which forms part of the skull table and a ventral process, which separates the orbit and infratemporal fenestra. The dorsal portion is a flat, mediolaterally expanded element which forms the lateral portion of the supratemporal fenestra. The medial side of the postorbital bears a supratemporal fossa that is continuous with the fossa of the frontal. This is also present in *Batrachotomus kupferzellensis* (SMNS 80260) and in *Postosuchus kirkpatricki* (Weinbaum, 2011). The fossa shallows posteriorly and disappears at the posterior portion. A posterolaterally directed ridge originates at the border of the supratemporal fenestra and crosses the postorbital to terminate on the lateral edge of both *Heptasuchus clarki* and *Batrachotomus kupferzellensis* (SMNS 80260). The posterior portion of the postorbitals of *Heptasuchus clarki*, *Batrachotomus kupferzellensis* (SMNS 80260), and *Postosuchus kirkpatricki* (TTUP 9000) are relatively wider than that of *Saurosuchus galilei* (PVSJ 32), a skull assigned to *Prestosuchus chiniquensis* (UFRGS T-156), and *Luperosuchus fractus* (UNLR 04). The posterior portion of the postorbital of *Heptasuchus clarki* appears to overlay the squamosal as in *Batrachotomus kupferzellensis* (SMNS 80260), and *Postosuchus kirkpatricki* (TTUP 9000) (see character 428).

The laterally oriented, rugose ridge continues from the postfrontal to the postorbital. The ridge splits into ventral and posterior components, with a small gap on the anterior side where the ridges come together (Fig. 6C). The ventral ridge forms the posterior margin of the orbit for the length of the ventral process. Directed ventrally at its origin, the ridge, along with the ventral process, curves gradually anteroventrally creating an arc of nearly 50°. The ridge is rugose and similar to that of *Batrachotomus kupferzellensis* (SMNS 80260) although the degree of rugosity differs among *Batrachotomus kupferzellensis* individuals (SMNS 80260 versus SMNS 52970). Posterior to the dorsal portion of the ridge, a large fossa is present that is roofed by the dorsal portion of the postorbital. This deep fossa is also present in *Batrachotomus kupferzellensis* (SMNS 80260) and also, to a lesser degree in *Saurosuchus galilei* (PVSJ 32), a skull assigned to *Prestosuchus chiniquensis* (UFRGS T-156), and *Postosuchus kirkpatricki* (TTUP 9000). The ridge terminates dorsoventrally in a broad flange that clearly entered the orbit and contributed to the 'keyhole shape' of the orbit. Additionally, a deep fossa is present on the anterodorsal side of the ventral termination of the postorbital. This deep fossa, which extends dorsally into the ventral process, is only visible in anterior view. A similar feature is also present in *Batrachotomus kupferzellensis* (SMNS 80260) and was originally considered

to be an autapomorphy of the taxon by *Gower (1999)* (see character 428). However, the fossa in *Batrachotomus kupferzellensis* is located only on the lateral surface whereas the feature in *Heptasuchus clarki* is only on the anterodorsal surface. It is not clear if this difference is the result of crushing in *Heptasuchus clarki*. Moreover, the depth of the fossa differs among *Batrachotomus kupferzellensis* individuals (SMNS 80260 versus SMNS 52970).

The ventral process of the postorbital is subrectangular in cross-section for the length of the element. The ventral process lacks the 'kink' as seen in *Batrachotomus kupferzellensis* (SMNS 80260), *Postosuchus kirkpatricki* (TTUP 9000), and *Saurosuchus galilei* (PVSJ 32). However, this 'kink' is subtle in taxa with the feature and may be difficult to detect if parts of the posteroventral margin of the ventral process are incomplete. In medial view, a shallow and broad groove posterior to a ridge on the anterior edge of the ventral process marks the articulation with the dorsal process of the jugal. The articular surface with the jugal is restricted to the posteroventral side of the ventral process. A shallow fossa is present at the dorsal margin of the ventral process and may represent the articular surface with the laterosphenoid.

**Parietal:** Only the lateral portion of the occipital process of the right parietal is preserved (Fig. 7). The process remains in articulation with the supraoccipital and possibly touches the paroccipital process posterior laterally. The vertically oriented process forms the dorsal portion of a large post temporal fenestra. A distinct ridge is present on the anterior side of the lateral process.

**Occiput and Braincase:** The three dimensionally preserved braincase (UW 11562-H) is largely complete on the right side and preserves the opisthotic, exoccipital, occipital and parabasisphenoid, prootic, and the right half of the supraoccipital (Fig. 7). The braincase elements are still in articulation with the occipital process of the right parietal. The bone surface is well preserved and details of the morphology of the medial surfaces are readily apparent. The braincase is well ossified and sutures between most elements cannot be distinguished in most cases.

The basioccipital forms the majority of the occipital condyle and the exoccipitals are completely fused to the dorsolateral surfaces (Fig. 7). A small notochordal pit is present on the dorsal portion of the basioccipital. The condylar stalk (=neck) is well expanded and a distinct rim outlines the circumference of the basioccipital. The preserved portion of the foramen magnum is semicircular in shape and its flattened floor extends onto the dorsal surface of the occipital condyle. The basitubera originate at the ventral portion of the occipital condyle and stretch ventrolaterally. As with *Batrachotomus kupferzellensis* (SMNS 80260), the basitubera are bilobed and are separated from the basitubera of the parabasisphenoid by an unossified gap. The unossified gap of *Heptasuchus clarki* is large like that of *Saurosuchus galilei* (PVSJ 32). The lateral edge of the more lateral lobe of the basitubera is continuous with the lateral ridge (sensu *Gower, 2002*) that originates on the exoccipital. The more medial lobe of *Heptasuchus clarki* is larger and is distinctly

convex in contrast to that of the basitubera of *Postosuchus kirkpatricki* (TTUP 9000). There is no division between the basioccipital and the parabasisphenoid at the midline.

Only the right exoccipital is fully preserved (Fig. 7). The exoccipitals meet on the midline similarly to most pseudosuchians other than crocodylomorphs and shuvosaurids (*Nesbitt, 2011*). The lateral side of the exoccipital bears a lateral ridge that obstructs the descending process of the opisthotic in posterior view, similar to that of *Batrachotomus kupferzellensis* (SMNS 80260), *Postosuchus kirkpatricki* (*Weinbaum, 2011*), crocodylomorphs and aetosaurs (*Gower & Walker, 2002*). Two foramina, interpreted as the exits of cranial nerve XII, pierce the medial surface of the exoccipital. However, only one exit cranial nerve XII can be observed on the lateral side of the exoccipital. This exit is located anterior to the lateral ridge and directed into the opening for the metotic opening as with *Batrachotomus kupferzellensis* (*Gower, 2002*). The opisthotic is fused with the exoccipital.

The well-preserved prootic, which separates the parabasisphenoid from the laterosphenoid, is complete (Fig. 7). However, the sutures with the surrounding elements are difficult to discern. The anterolateral surface bears the exits for cranial nerves V and VII. The exit for cranial nerve V appears to lie completely within the prootic as in *Postosuchus kirkpatricki* and Postosuchus alisonae (*Weinbaum, 2011*; *Peyer et al., 2008*) and not shared with the laterosphenoid as in *Batrachotomus kupferzellensis* (*Gower, 2002*) and *Sphenosuchus acutus* (*Walker, 1990*). A fossa surrounds the opening for cranial nerve V in *Heptasuchus clarki*. Anteriorly, a groove is present linking the exit for cranial nerve V and the anterior edge. A notch on the anterodorsal edge, just anteromedial to the exit of cranial nerve V, possibly represents the exit of the middle cerebral vein. A slight groove leads anteriorly into this notch. A small ridge located dorsal to the exit of cranial nerve V is interpreted to be the site of attachment for the protractor pterygoidei following *Gower & Sennikov (1996)* and *Gower (2002)*. There is a vertical ridge on the small anterior portion of the prootic just anteroventral to the exit of cranial nerve V. The pathway of cranial nerve IV appears to pierce the anterior, upturned process of the prootic. This process separates the laterosphenoid from the parabasisphenoid.

The exit for cranial nerve VII is located in a posterolaterally opening slot on the posterolateral portion of the prootic (Figs. 7A and 7B). The deep pocket for the exit of cranial nerve VII continues ventrally as a groove on the lateral side of the parabasisphenoid. The surface between the exits of cranial nerves V and VII is concave. There is no articular surface on the anterolateral surface of the prootic and the quadrate head as in crocodylomorphs (*Gower, 2002*).

Medially, the surface of the prootic is not well preserved. There is no clear pneumatization of inner ear as in crocodylomorphs as described by *Walker (1990)*. The medial wall of the vestibule appears to be nearly fully ossified as with most suchians (*Gower, 2002*; *Gower & Nesbitt, 2006*), but the center of the wall is broken.

The right opisthotic is completely preserved (Fig. 7). The stapedial groove leading into the fenestra ovalis is shallow and poorly defined anteriorly. The descending process of the opisthotic (=crista interfenestralis) divides the metotic foramen anteriorly from the fenestra ovalis posteriorly. This thin process of the opisthotic is expanded mediolaterally.
Nearly all of the descending process of the opisthotic is hidden posteriorly by the lateral ridge on the exoccipital in *Heptasuchus clarki*, as in aetosaurs, *Batrachotomus kupferzellensis, Postosuchus kirkpatricki*, and crocodylomorphs (*Gower, 2002*). There does not appear to be a foramen in the dorsal portion of the metotic opening as there is in *Batrachotomus kupferzellensis* (*Gower, 2002*), but this area is incompletely prepared. The perilymphatic foramen is not fully ossified, but must have been oriented posteriorly and not laterally as in *Sphenosuchus acutus* (*Walker, 1990*) and other crocodylomorphs (*Gower, 2002*).

Lateral to the foramen magnum, the paroccipital processes of the opisthotics are constricted (to 2.3 cm) at their bases but broaden considerably (to 5.2 cm) to form club-shaped posterolateral expansions (Fig. 7). The processes are directed dorsolaterally at an angle of 35° from the vertical plane of the occiput. The broadness of the lateral portions of the paroccipital processes is greater than that of *Batrachotomus kupferzellensis* (SMNS 80260), but similar to *Postosuchus kirkpatricki* (*Weinbaum, 2011*) and crocodylomorphs (e.g., *Sphenosuchus acutus*). The ventral portion of the process of *Heptasuchus clarki* is nearly straight whereas the dorsal margin is significantly expanded dorsally. The dorsal edge of the process forms the ventral margin of a clear post temporal fenestra. Shallow grooves are present on the ventral surface of the paroccipital process. The lateral edge of the paroccipital is rounded like that of *Batrachotomus kupferzellensis* (SMNS 80260).

The basisphenoid and parasphenoid are fused together to form a parabasisphenoid. The body of the parabasisphenoid is vertically oriented where the basipterygoid processes are extended well ventral of the basitubera. The parabasisphenoid portion of the basitubera project laterally and dorsolaterally at its tips. A deep fossa (=medial pharyngeal recess, =hemispherical fontanelle) is positioned between the basitubera and the midline. This depression is undivided on the midline, whereas there is a distinct lamina of bone dividing the depression in *Batrachotomus kupferzellensis* (*Gower, 2002*) and *Sphenosuchus acutus* (*Walker, 1990*). There is no intertubural plate (*Gower & Sennikov, 1996*) across the midline. The body of the parabasisphenoid is waisted between the basitubera and the basipterygoid processes. The posteriorly directed basipterygoid processes extend ventrally beyond the rest of the braincase. The articular surfaces with the pterygoid are positioned on the anterior portion of the basipterygoid processes. The posterior portions of the processes expand posterodorsally into mediolaterally thin sheets of bone. These processes are autapomorphic (see diagnosis) for *Heptasuchus clarki* and represent a clear difference between *Heptasuchus clarki* and *Batrachotomus kupferzellensis*.

Laterally, the entrance of the internal carotid arteries lies in the groove that is continued from the prootic on the lateral side of the parabasisphenoid (Fig. 7). The path of the internal carotid travels anteriorly to exit at the base of the hypophyseal fossa as observed on the broken left lateral side. The articulation of the descending process of the opisthotic with the parabasisphenoid is not distinct. The base of both the metotic fenestra and the fenestra ovalis are broadly rounded and lie on the dorsal portion of the parabasisphenoid. The ventral base of the metotic fenestra is well ventral to the contact between the basioccipital and the exoccipital.

The cultriform process is complete, relatively short compared with the braincase, and dorsoventrally expanded posterior to the anteriorly tapering tip (Fig. 7). A dorsoventrally expanded cultriform process is also present in *Batrachotomus kupferzellensis* (*Gower, 2002*) and *Postosuchus kirkpatricki* (*Weinbaum, 2011*). A distinct ventral step is present in the anterior half of the element. There does not appear to be a longitudinal groove on the dorsal surface of the cultriform process as there is in *Arizonasaurus babbitti* (*Gower & Nesbitt, 2006*). Comparisons with the length and dorsoventral depth of the cultriform process are limited among suchians given that this region is not common preserved.

Dorsal to the foramen magnum, the vertically inclined face of the supraoccipital extends dorsally to contact the parietal.

**Quadrate:** The dorsal (UW 11563-AD) and ventral portions (UW 11563-AF, UW 11563-H) of the left quadrate were found among the weathered elements collected at the locality. The dorsal fragment (Fig. 8D) that articulated with the squamosal, is rounded in dorsal view, and the surface is composed of spongy bone circumscribed by a ring of compact bone. There is no posterior hook of the quadrate as there is in *Postosuchus kirkpatricki* (TTUP 9000). The ventral portion consists of the articular facet with the articular (Figs. 8A–8C). The convex facet is divided into medial and lateral condyles separated by a shallow fossa. The more medial condyle of the articular surface projects further ventrally than the lateral condyle. The ventral articular surfaces lap dorsally onto the anterior surface. Anteriorly, a small but well-defined ridge originates on the lateral condyle and trends dorsomedially.

**Palate:** A nearly complete left palatine (UW 11562-K, Fig. 6F) is represented in the type specimen. The thin medial, anterior and posterior portions of the element are incomplete. The body of the palatine is thin for most of the length of the element. The lateral side bears a dorsoventrally expanded, anteroposteriorly straight facet for articulation with the medial side of the maxilla. In dorsal view, the expansion forms a lateral lip on the lateral side of the element. The posterolateral portion forms the anteromedial margin of the suborbital fenestra and the posterior portion tapers posteromedially. Anteriorly, only a portion of the dorsal fossa that holds the pterygoideus muscle (*Witmer, 1997*) is preserved. The portion preserved suggests that the fossa is anteriorly shifted near the choana as in *Batrachotomus kupferzellensis* (*Gower, 2002*) relative to the more posterior position in *Polonosuchus silesiacus* (ZPAL Ab/III 563), *Saurosuchus galilei* (PVSJ 32), aetosaurs (*Gower & Walker, 2002*), and the crocodylomorph *Sphenosuchus acutus* (*Walker, 1990*). The posterior border of the choana is thickened relative to the body in *Heptasuchus clarki* but does not possess a surrounding rim in the same area as in *Polonosuchus silesiacus* (ZPAL Ab/III 563). Ventrally, the surface is nearly flat except for a shallow facet for the articulation with the pterygoid on the posteromedial portion.

**Pterygoid:** Two elements (UW 11562-L and UW 11562-M; Figs. 8E–8H) not readily identified originally were found in situ with the holotype; here we interpret these fragments as parts of the pterygoid. UW 11562-L consists of a thin, plate like element that is possibly part of the lateral process of the pterygoid. All sides except one, presumably the medial side, are broken. The 'medial' side is straight with a distinct step at the edge near the middle

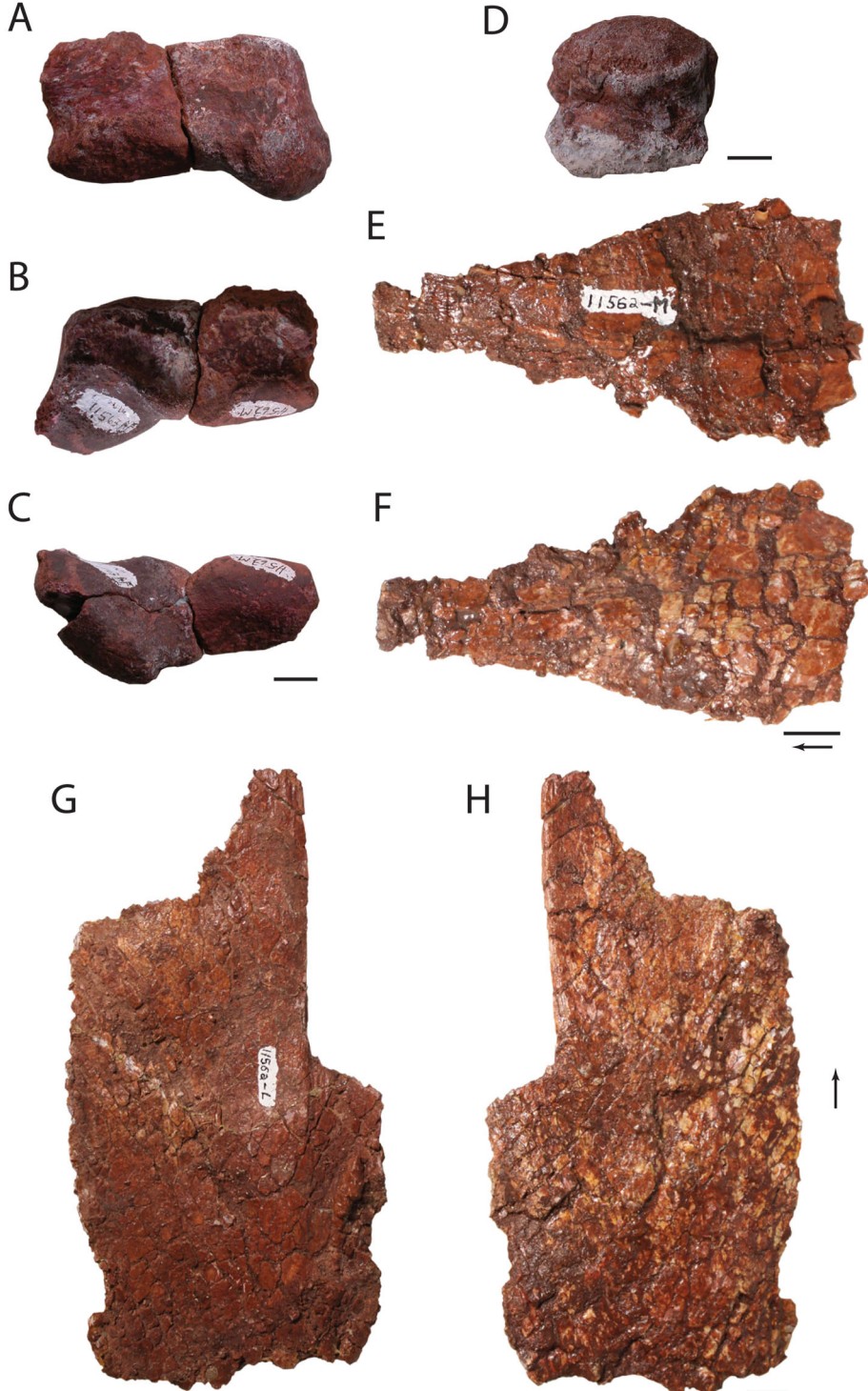

**Figure 8 Fragmentary skull elements of *Heptasuchus clarki*.** Ventral portion of the left quadrate (UW 11563-AF + UW 11563-H, labeled before putting together) in posterior (A), anterior (B), and ventral (C) views; dorsal head of the quadrate (side unknown; UW 11562) in lateral? (D) view; possible fragments of the pterygoid (UW 11562-M) in two (E–F) views; possible fragment of the pterygoid (UW 11562-L) in two (G–H) views. Arrows indicate anterior direction. Scales = 1 cm.

of the element. Here the bone is rugose and may serve as an articular facet. The essentially flat surfaces are nearly featureless. UW 11562-M is a thin fragment that may pertain to the anterior (=palatine) process of the pterygoid. The element likely tapers anteriorly and between longitudinal ridges on both sides.

**Dentition:** A single premaxillary tooth (UW 11562-A), the first five teeth of the left maxilla (UW 11562-C) and the fourth, sixth, and ninth tooth of the right maxilla (UW 11562-B) are preserved in place in the holotype (Fig. 5). Loose teeth (UW 11562-AA through -AI) found at the locality are referred to *Heptasuchus clarki* based on similarity, but only the teeth found in the tooth bearing bones are described in detail. The roots of the premaxillary and maxillary teeth lie in deep sockets.

The only preserved premaxillary tooth, in either tooth position two or three (Figs. 5E and 5F), is unique among the other teeth preserved in *Heptasuchus clarki* in that its crown is cylindrical in shape and bears no serrations. The tip grades into a distal portion, which is laterally compressed to form a blade similar in shape to the distal tips of the maxillary teeth. The axis of this blade, however, lies at an angle to the blade axis of the maxillary teeth.

Generally, the maxillary teeth are ziphodont in that they are mediolaterally compressed, recurved, and bear serrations on the mesial and distal sides. The crowns are long, that of a fully erupted tooth being approximately equal in length to its root. Typically, there are 12 serrations per 5 mm. The left maxilla (Figs. 5A and 5B), bearing the first five teeth of the maxillary series, clearly shows the pattern of tooth replacement. As in *Saurosuchus galilei* (*Sill, 1974*), the teeth grow and are replaced in two alternating waves. Teeth in positions three and five were newly erupted when the individual was buried whereas teeth in positions two and four are fully erupted. Tooth position two shows especially severe signs of wear, as its tip is badly blunted and the serrations were worn away, likely in life. The right maxilla (Figs. 5C and 5D), with the medial wall almost entirely removed by erosion also illustrates the process of tooth replacement in *Heptasuchus clarki*; tooth position six is fully erupted and a replacement tooth lies on its lingual surface within a socket of the fully erupted tooth at the base of its root.

### Postcranial Skeleton

The postcranial of *Heptasuchus clarki* is only represented by a few complete or nearly complete bones (e.g., pubis, tibia, ulna) whereas most other postcranial elements were found on the surface after extensive surface weathering. It is apparent that much of the shaft of limb bones and delicate parts of vertebrae (e.g., base of the neural arches) were weathered away much more easily than the more robust elements, such as limb bone ends and centra fragments. A few postcranial bones were found in place (e.g., trunk vertebra; Figs. 9A and 9B), but suffer from poor surface details with few exceptions.

**Vertebrae:** The vertebral column of *H. clarki* is represented by only a few poorly preserved centra, one complete neural spine, and a large number of fragments from neural arches (e.g., diapophyses from trunk vertebrae) along the column. Those centra that are

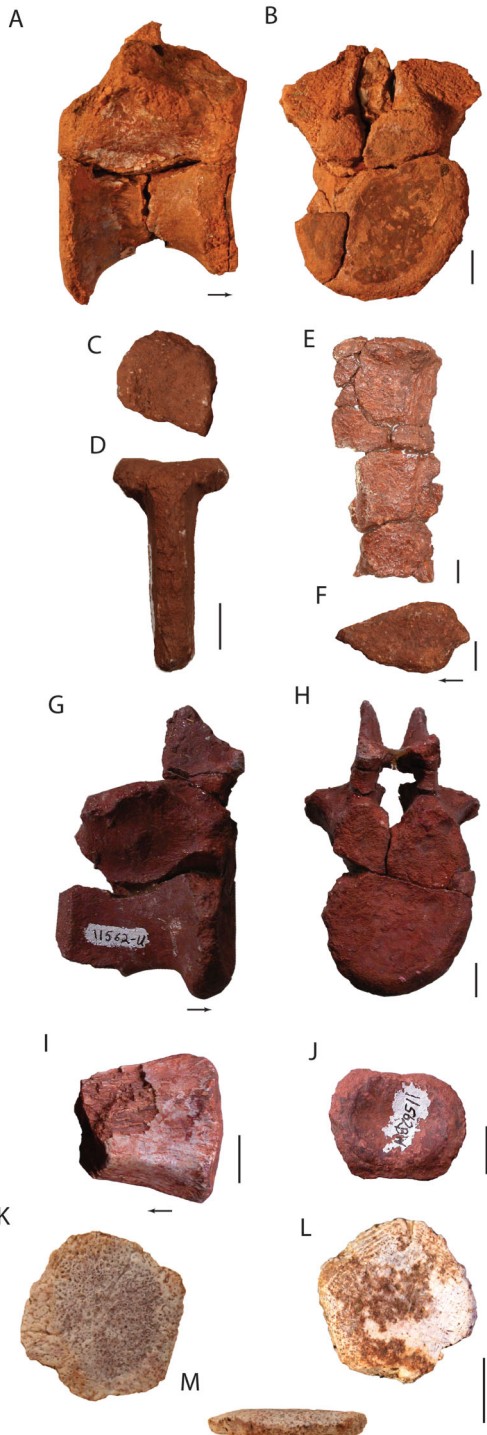

**Figure 9 Axial elements of *Heptasuchus clarki*.** Posterior trunk vertebra (TMM 45902-2) in right lateral (A) and posterior (B) views; neural spine of a cervical-trunk vertebra (UW 11562-CX) in dorsal (C) and posterior (D) views; presacral neural spine (UW 11562-V) in lateral (E) view; presacral neural spine (UW 11562-CT) in lateral (F) view; anterior caudal vertebra in lateral (G) and anterior (H) views; distal caudal vertebra (UW 11562-BW) in ventral (I) and posterior (J) views; osteoderm (TMM 45902-1) in three views; anterior caudal vertebra in dorsal (K), ventral (L), and lateral (M) views. Arrows indicate anterior direction. Scales = 1 cm.     

sufficiently preserved to warrant description include parts of three cervicals, a trunk, and parts of caudal centra.

The most anterior vertebra represented among the referred material consists of a fragmentary centrum (UW 11562-T) from approximately the middle of the cervical series (comparing to that of *Postosuchus kirkpatricki* *Weinbaum, 2013*) which retains the anterior and posterior articular surfaces and the length of this centum is a bit less than its height, typical of loricatan taxa with short necks (e.g., *Batrachotomus kupferzellensis*, *Gower & Schoch, 2009*; *Postosuchus alisonae*, *Peyer et al., 2008*; *Prestosuchus chiniquensis*, *Desojo, Baczko & Rauhut, 2020*). Between the articular faces, the centrum is constricted in ventral view. Lateral to the anterior articular facet, the parapophyses sit on the ventral half of the centrum and project laterally. They are separated by a ventrally projecting lip, which originates from the ventral portion of the anterior facet. The ventral surface of the centrum bears a slight ridge (=keel), as typical of most archosauriforms (*Nesbitt, 2011*).

A more posterior cervical centrum is represented by just the anterior portion (UW 11564-A). The anterior articular facet is circular and only slightly concave/amphicoelous. Lateral to the anterior articular facet, the parapophyses lie slightly more dorsally on the centrum than in UW 11562-T. The parapophyses face laterally with a slight posterior component. Just posterior to the anterior articular facet, the centrum constricts rapidly to the point where it has broken, preserving only about half of the total length of the element based on our estimation and comparisons to *Batrachotomus kupferzellensis* (*Gower & Schoch, 2009*) and *Postosuchus alisonae* (*Peyer et al., 2008*). The marked constriction decreases width from 4.5 cm at the anterior articular facet rim to l.5 cm at the midpoint. A trace of a faint ridge (=keel) is present on the midline of the ventral surface. In this vertebra, as in all those preserved in *Heptasuchus clarki*, the neural canal deeply indents the dorsal portion of the body of the centrum behind the flared rim. This condition "central excavation" is present in archosauriforms outside crown Archosauria *Euparkeria capensis* (Ewer l965), and also within the crown group (e.g., *Arizonasaurus babbitti*, *Nesbitt, 2005*).

A nearly complete trunk centrum (TMM 45902-2; Figs. 9A and 9B) was excavated from the ground in 2009, but the specimen is poorly preserved and lacks the process of the neural arch. TMM 45902-2 likely represents a mid to posterior trunk vertebra based on the dorsal and posteriorly placed parapophysis based on comparison with other loricatans (e.g., *Batrachotomus kupferzellensis*, *Gower & Schoch, 2009*). The anterior and posterior articular facets of the centrum are nearly circular, with a slightly taller dorsoventral height compared to the mediolateral width. The centrum rims are well pronounced, but slightly weathered, and the centrum is well constricted in both lateral and ventral views between the articular facets. The neurocentral suture is fused and no trace of the suture can be observed. The lateral portions of the diapophyses are broken, but the base is shifted posteriorly and likely connected with the base of the diapophyses. Posteriorly, the neural canal is oval, with a much greater height dorsoventrally than mediolateral width. This width to height to ratio of 0.7 in *Heptasuchus clarki* is much higher than in closely related taxa (e.g., *Batrachotomus kupferzellensis*, *Gower & Schoch, 2009*; *Postosuchus alisonae*, *Peyer et al., 2008*; *Stagonosuchus nyassicus*, *Gebauer, 2004*).

A mostly complete caudal vertebra (UW 11562-U; Figs. 9G and 9H) comprises a nearly complete centrum and part of the neural arch. We interpret this as a more anterior caudal vertebra given that the centrum is about as tall as long, lacks any clear facets for the chevron, and the transverse processes, although broken, are large and similar to those of the anterior caudal vertebrae of *Prestosuchus chiniquensis* (SNSB-BSPG AS XXV 3b; *Desojo, Baczko & Rauhut, 2020*). The anterior articular facet of the centrum (Fig. 9F) is ellipsoidal with a dorsoventral height of five centimeters compared to a mediolateral width of four centimeters. Additionally, the anterior articular facet is slightly concave, like the other vertebrae throughout the column. The centrum is constricted just posterior to the well-defined rim of the anterior articular facet. Only a small fraction of the posterior articular facet is preserved. The anterior portion of the neural arch is intact, including bases of the prezygapophyses. The articular facets of the prezygapophyses are low, ~20° to the horizontal. Dorsal to the neural canal, the beginnings of the neural spine project dorsally, flanking a deep interspinous cleft (Fig. 9G) as in *Saurosuchus galilei* (*Sill, 1974*). As in the other vertebrae described, the neural canal expands ventrally into the dorsal surface of the centrum.

A number of partial centra of distal caudal vertebrae are preserved (UW 11563-A-C; UW 11562-BW; Figs. 9I and 9J); none preserve the neural spine. The posterior caudal vertebrae are typical of archosaurs (e.g., *Postosuchus alisonae*; NCSM 13731) in that the centra are longer than tall, lack lateral processes, and the middle of the centrum is only slightly constricted relative to the articular facets. The width of the centra (Figs. 9G and 9H) is similar to those of *Postosuchus kirkpatricki* (TTUP 9002), but do not appear to be unique among archosaurs given the paucity of posterior caudal vertebrae associated with diagnostic material.

A number of neural spines were found among the surface collected material, but the exact position of each neural spine within the vertebral column cannot be reconstructed precisely. The height of the neural spines are difficult to estimate, but most of a neural spine (UW 11562-V; Fig. 9E) shows that at least some of the neural spines were about twice the height of a trunk centrum. The neural spines are blade-like in anterior and posterior views and clearly bear lateral expansions at the dorsal end of the spine. The lateral expansions are globular in lateral view and obtain their greatest lateral expansion near the anteroposterior center (UW 11562-CT) or slightly posterior to the anteroposterior center. Additionally, the lateral expansions appear to not expand anteriorly or posteriorly compared to the rest of the neural spine. There is clear variation in the sample; the lateral expansions are greater in some specimens (UW 11562-CT) compared to others (UW 11562-V). In dorsal view, some appear nearly circular (UW 11562-CX) whereas others are more 'heart-shaped' with a posterior prong present at the midline (UW 11562-CT). These expansions, referred to as spine tables by some authors (e.g., see *Nesbitt, 2011*), commonly occur in non-crocodylomorph loricatans such as *Batrachotomus kupferzellensis* (*Gower & Schoch, 2009*), *Stagonosuchus nyassicus*, (*Gebauer, 2004*), *Saurosuchus galilei* (*Trotteyn, Desojo & Alcober, 2011*), *Prestosuchus chiniquensis* (ULBRA-PVT-281; *Da Silva et al., 2018*), and in the cervical vertebrae of *Postosuchus kirkpatricki* (*Weinbaum, 2013*), and clearly outside the group (e.g., *Nundasuchus songeaensis Nesbitt et al., 2014*).

The morphology of the lateral expansions of the dorsal portion of the neural spines are abundant enough to support that both the cervical and the trunk vertebrae had the feature, as in *Batrachotomus kupferzellensis* (*Gower & Schoch, 2009*).

**Osteoderm:** A single osteoderm (Figs. 9K–9M) was recovered among the holotype in 2010 (TMM 45902-1). The size of the osteoderm is consistent with that of *Heptasuchus clarki*, but it is impossible to conclude that the osteoderm definitely belonged to *Heptasuchus clarki*. The semicircular osteoderm has a nearly flat outer surface covered in small foramina and a few short canals connecting some of the foramina. The ventral surface is nearly smooth with small crisscrossing bone fibers as in most archosauriform osteoderms. In lateral view, the osteoderm is compressed and dorsal and ventral sides are parallel for much of their length, both sides taper toward the edges. The location of the osteoderm on the skeleton is not known and there is no anterior process is present as in most pseudosuchians (*Nesbitt, 2011*).

**Scapula**: Two partial scapulae, consisting solely of the glenoid region, are known from the accumulation. The larger specimen (UW 11566-B) and smaller specimen (UW 11565-E; Fig. 10A) is from the right side. The larger specimen indicates that the coracoid may be partially coossified to the scapula whereas the smaller specimen clearly has a contact surface with the coracoid. The glenoid is well defined by a rim and the glenoid itself is weakly concave. The glenoid opens posteriorly with a lateral component, but the exact angle cannot be determined because the rest of the scapula is not present; the orientation of what is preserved is similar to that of *Batrachotomus kupferzellensis* (SMNS 80271). Just distal to the glenoid on the posterior edge, a rugose scar marks the surface for origin of M. triceps as in other archosaurs (*Gower & Schoch, 2009*). This scar is rugose and distinct in *Heptasuchus clarki*, but not nearly as laterally expanded compared to that of *Batrachotomus kupferzellensis* (SMNS 80271).

**Coracoid:** Two fragmentary coracoids (UW 11566; Fig. 10B) were recovered as float during the initial excavation. Both coracoids consist of the more robust glenoid region with a broad articulation surface with the scapula. The laterally concave articulation surface with the humerus (=glenoid) project posterolaterally like that of *Batrachotomus kupferzellensis* (SMNS 80271) and *Postosuchus kirkpatricki* (TTUP 9002). In proximal view, the rugose articulation surface with the scapula is triangular and extends laterally into a small peak. The anterolateral surface just distal to this articulation surface is striated and flat. A clear coracoid foramen is present anterior to the largest articulation surface with the scapula. The foramen is only partially complete in both specimens; but shows that the foramen nearly contacted the scapula articulation surface on the medial surface. The medial surface is flat. It is not clear if the coracoid of *Heptasuchus clarki* had a postglenoid process.

**Humerus:** A proximal portions of a left humerus (UW 11565-A; Figs. 10C and 10D) and the proximal portion of a second left humerus (UW 11563-U) are represented among the referred material of *Heptasuchus clarki*. The latter bone, collected outside the quadrant

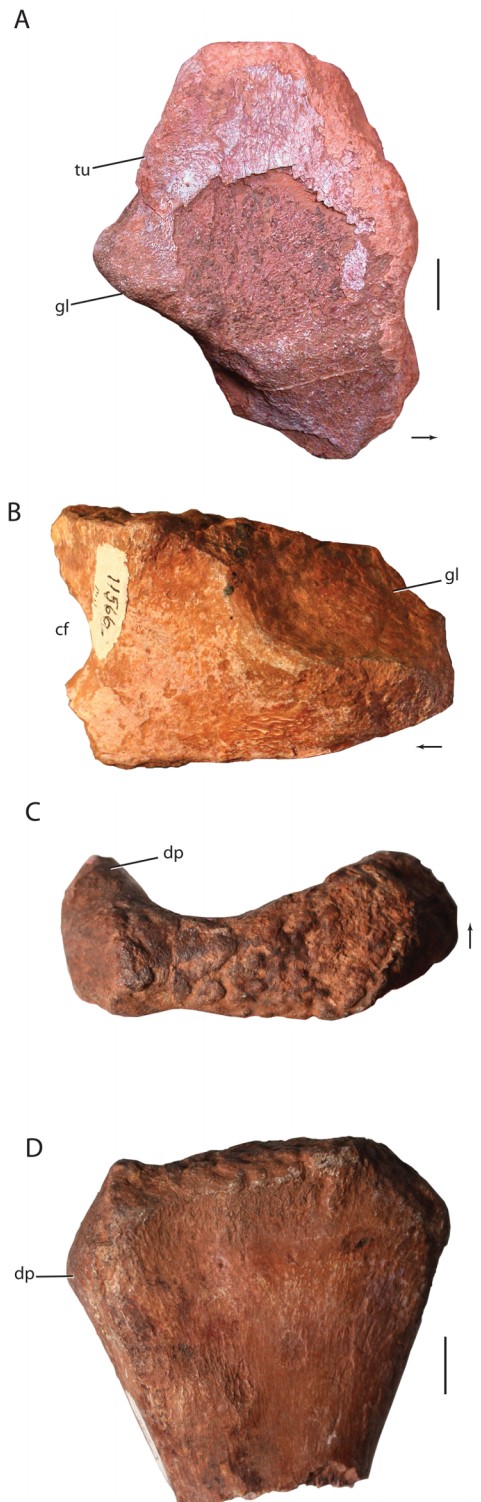

**Figure 10 Pectoral elements and incomplete humerus of *Heptasuchus clarki*.** Right incomplete scapula (UW 11565-E) in lateral (A) view; incomplete left coracoid (UW 11566) in lateral (B) view; proximal portion of left humerus (UW 11565-A) in proximal (C) and posterior (D) views. Arrows indicate anterior direction. Abbreviations: cf, coracoid foramen; dp, deltopectoral crest; gl, glenoid; tu, tuber. Scales = 1 cm.
system, is weathered, but clearly indicates the presence of a slightly smaller individual from the locality.

The surfaces of UW 11565-A are well preserved. The overall proportions of the humerus cannot be specifically determined because the shaft and distal end are missing. However, it is clear that the proximal expansion relative to the shaft would have been less in *Heptasuchus* and other forms like *Batrachotomus kupferzellensis* (SMNS 80276), *Postosuchus kirkpatricki* (TTUP 9002), *Ticinosuchus ferox*, and crocodylomorphs rather than the largely expanded proximal portions of *Stagonosuchus nyassicus* (GPIT/RE/3832), and aetosaurs and their close relatives (e.g., *Parringtonia gracilis*, NMT RB426) where the medial and lateral edges diverge at a greater angle proximally. The proximal surface of the bone is rugose, possibly indicating that ossification of the proximal end was not complete at the time of death. The proximal surface lacks a rounded 'head,' as present in *Batrachotomus kupferzellensis* (SMNS 80276), *Postosuchus kirkpatricki* (TTUP 9002), and early crocodylomorphs (*Nesbitt, 2011*). In proximal view, the medial portion expands relative to the narrower middle to lateral portion. In posterior view, the medial portion of the proximal surface is rounded and is deflected distally. More laterally, the proximal surface bears a distinct peak near the origin of the deltopectoral crest. The distinct peak (Figs. 10C and 10D), which is best observed in posterodorsal view, occurs in *Batrachotomus kupferzellensis* (SMNS 80276) and *Stagonosuchus nyassicus* (GPIT/RE/3832), to a lesser extent in *Mandasuchus tanyauchen* (NHMUK PV R6793), but absent in *Postosuchus kirkpatricki* (TTUP 9002) and early crocodylomorphs (*Nesbitt, 2011*). Broken in UW 11565-A, the deltopectoral crest of UW 11563-U shows that the structure is continuous with the proximal surface, as in *Mandasuchus tanyauchen* (NHMUK PV R6793) and *Batrachotomus kupferzellensis* (SMNS 80276) and not distally shifted as in *Postosuchus kirkpatricki* (TTUP 9002), and early crocodylomorphs (*Nesbitt, 2011*). The apex of the deltopectoral crest, which is triangular in lateral view, is located in a similar position as in *Batrachotomus kupferzellensis* (SMNS 80276). The anterior surface of the proximal portion is concave whereas the posterior surface is nearly flat. A weakly defined scar is present on the posterolateral side of the posterior surface and is equivalent to a scar in *Batrachotomus kupferzellensis* (SMNS 80276), interpreted to be the surface for origin of M. triceps (*Gower & Schoch, 2009*).

**Ulna:** A complete right ulna (UW 11562-W) and a nearly complete left ulna (UW 11562-X) are included as referred specimens (Figs. 11I–11L). Additionally, the distal ends of two other ulnae (UW 11563-V and UW 11565-C) are present indicating that at least three individuals were buried together at the locality. UW 11562-W measures 23.5 cm long and is nearly as long as the complete tibia (UW 11562-Z), but the ulna has a much smaller radius throughout the shaft. The ulna has an expanded proximal portion relative to the shaft and the shaft narrows distally for 2/3rds the length of element and then slightly expands at the distal end (Fig. 11). The expanded proximal end of the ulna bears a moderately developed olecranon process as demonstrated by UW 11562-X (Figs. 11I–11L). It appears that the olecranon process of UW 11562-W was a separate ossification and was not fused onto the proximal surface at the time of death. Comparatively, the olecranon is relatively smaller in

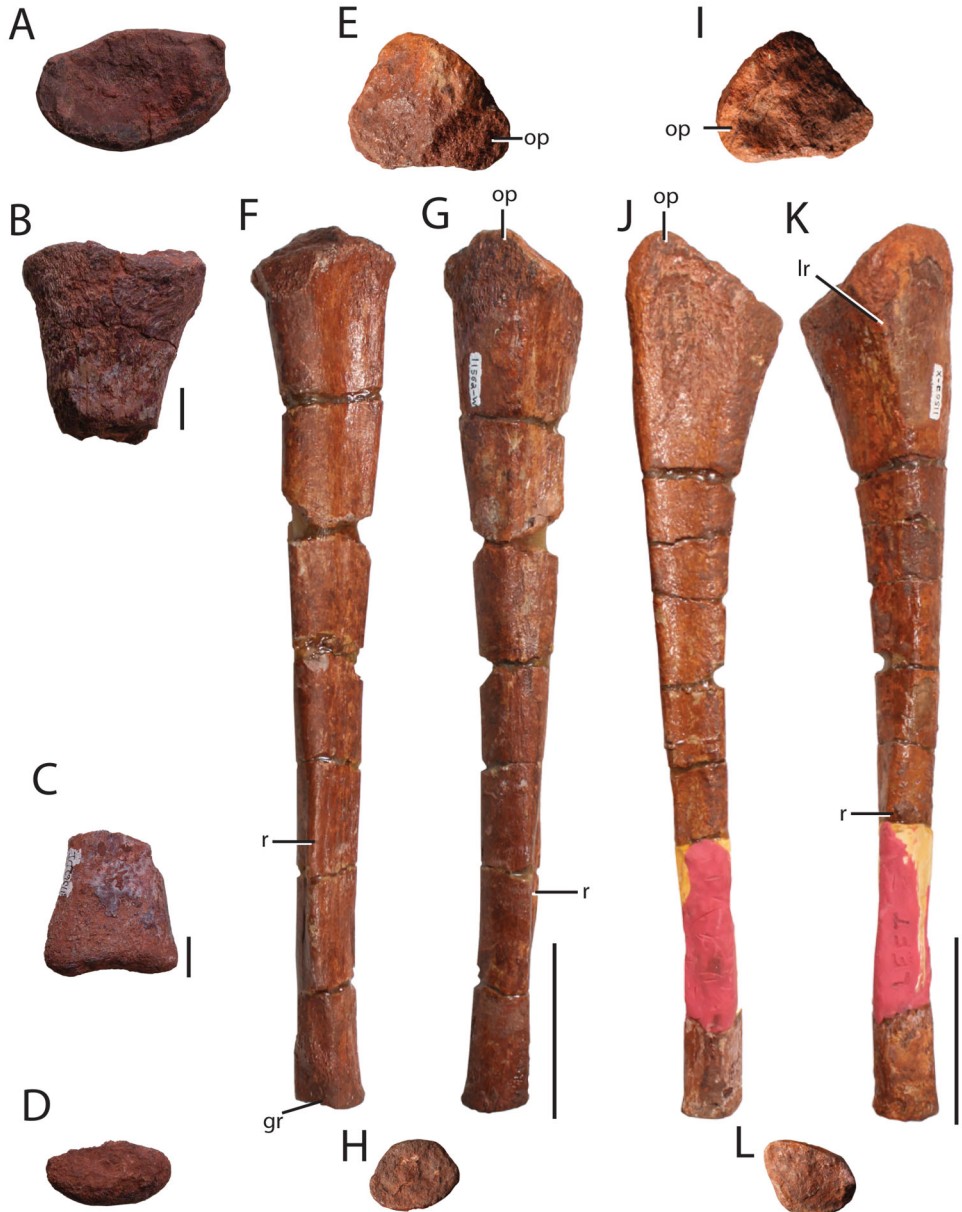

**Figure 11 Forelimb elements of *Heptasuchus clarki*.** Proximal portion of the radius (UW 11562-DM) in proximal (A), and lateral (B) views and the distal portion of the radius (UW 11562-DI) in ?anterior (C) and distal (D) views; right ulna (UW 11562-W) in proximal (E), medial (F), posterior (G), and distal (H) views; left ulna (UW 11562-X) in proximal (I), posterior (J), anterior (K), and distal (L) views. Abbreviations: gr, groove; lr, lateral ridge; op, olecranon process; r, ridge. Scales = 1 cm in (A–D) and 5 cm in (E–L).

*Heptasuchus clarki* than in aetosaurs (e.g., *Stagonolepis robertsoni*, Walker 1961), *Postosuchus kirkpatricki* (TTUP 9002), *Batrachotomus kupferzellensis* (SMNS 80275), and crocodylomorphs (e.g., *Hesperosuchus agilis*, Colbert, 1952) and is more similar in size to that of *Ticinosuchus ferox* (Krebs 1965) and *Mandasuchus tanyauchen* (NHMUK PV R6793). The proximal surface is rugose and triangular (Figs. 11E and 11I) with a distinct radial tuber, but this tuber is not as well expanded as that of *Postosuchus kirkpatricki* (TTUP 9000).

The radial tuber extends distally for about 1/3 the length of the ulna. The medial side of the proximal portion is concave, as in *Batrachotomus kupferzellensis* (SMNS 80275). The shaft of the ulna is circular, and the anterior surface of the bone bears a longitudinal ridge, that twists medially toward the distal end, where a narrow groove is formed between it and the medial edge of the bone. This ridge and groove appear to be present in both UW 11562-W and UW 11562-X and is autapomorphic for *Heptasuchus clarki* (see diagnosis). The rugose distal surface is ovoid in outline with a slightly tapered anterolateral end.

**Radius:** Only the ends of the radius have been identified from weathered fragments, but determining which side these elements are from is difficult. The proximal portion is represented by UW 11566-T and UW 11562-DM (Figs. 11A and 11B) and the possible distal ends are represented by UW 11562-DF and UW 11562-DI (Figs. 11C and 11D). The proximal end of the radius is mediolaterally compressed with anterior and posterior tapered ends. A concave surface, in lateral view, lies between the anterior and posterior ends of the proximal surface. The distal end is rounded anteriorly and possibly posteriorly also, but this cannot be confirmed because the posterior portion is broken.

**Ilium:** A fragment consisting of much of the pubic peduncle, and part of the acetabulum is the only positively recognized part of the of the ilium known (UW 11563-Y and UW 11563; Fig. 12F). In anteroventral view, the articulation surface with the pubis is rugose and triangular. The acetabular portion that is preserved is concave and the acetabulum appears to be imperforate, as expected for a non-crocodylomorph pseudosuchian. The surface within the acetabulum is smooth.

**Pubis:** A nearly complete left pubis (UW 11562-Y; Figs. 12A–12D) of *Heptasuchus clarki* was recovered; only parts of the thin medial portion of the pubic apron are not preserved. The pubis is ~37 cm in length from the articulation surface with the ilium to the distal surface. In lateral view, the bone is nearly straight along its entire length like that of *Batrachotomus kupferzellensis* (SMNS 80270). The proximal surface of the pubis articulates with the pubic peduncle of the ilium dorsally and ventrally, the proximal portion of the pubis contributes only a minor portion of the edge of the acetabulum, as in *Saurosuchus galilei* (Sill, 1974). Distally, the proximal portion narrows in lateral view and transitions into the shaft laterally and medially with the pubic apron. The lateral surface of the proximal portion bears a fossa surrounded by a rugose surface, as in *Batrachotomus kupferzellensis* (SMNS 80270); this surface marks the hypothesized site of origin of the *M. ambiens* (Gower & Schoch, 2009). Medially, the proximal portion of the apron is broken so that that the exact size of the obturator foramen cannot be determined, but it appears to be small like that of *Batrachotomus kupferzellensis* (Gower & Schoch, 2009), rather than the larger opening in *Postosuchus kirkpatricki* (Weinbaum, 2013). The anteroposteriorly thickened medial process marks the proximal articulation with its antimere as in nearly all paracrocodylomorphs.

In posterior and anterior views, the shaft bows laterally (Fig. 12B) and a similar morphology is absent in other paracrocodylomorphs. The shaft is rounded laterally and

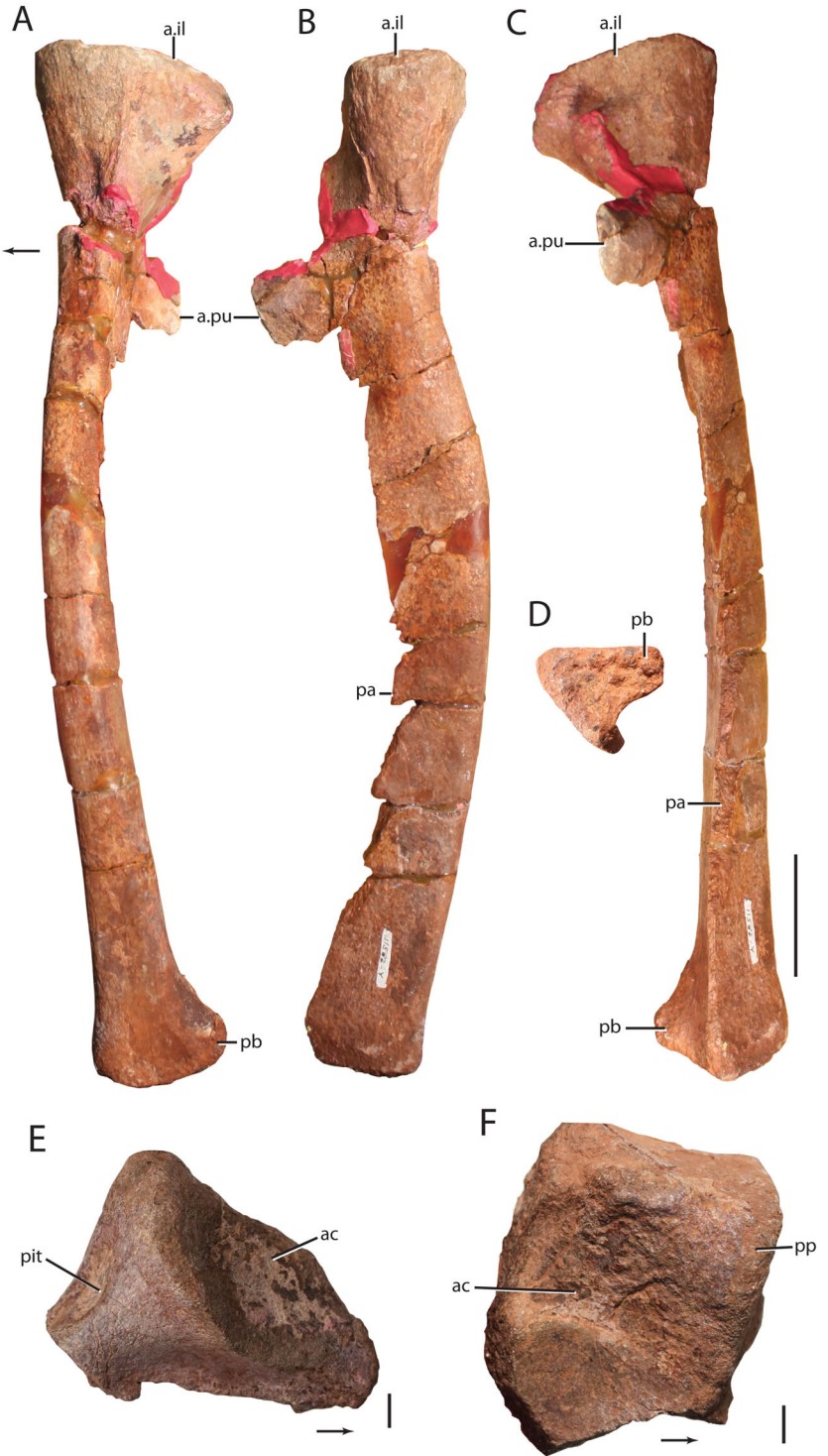

**Figure 12** **Pelvic elements of *Heptasuchus clarki*.** Left pubis (UW 11562-Y) in lateral (A), anterior (B), medial (C), and distal (D) views; proximal portion of the right ischium (UW 11564-B) in lateral (E) view; pubic peduncle of the right ilium (UW 11563) in lateral (F) view. Abbreviations: a., articulates with; as, acetabulum; il, ilium; pa, pubic apron; pb, pubic boot; pit, pit; pp, pubic peduncle; pu, pubis. Arrows indicate anterior direction. Scales = 5 cm in (A–B) and 1 cm in (E–F).

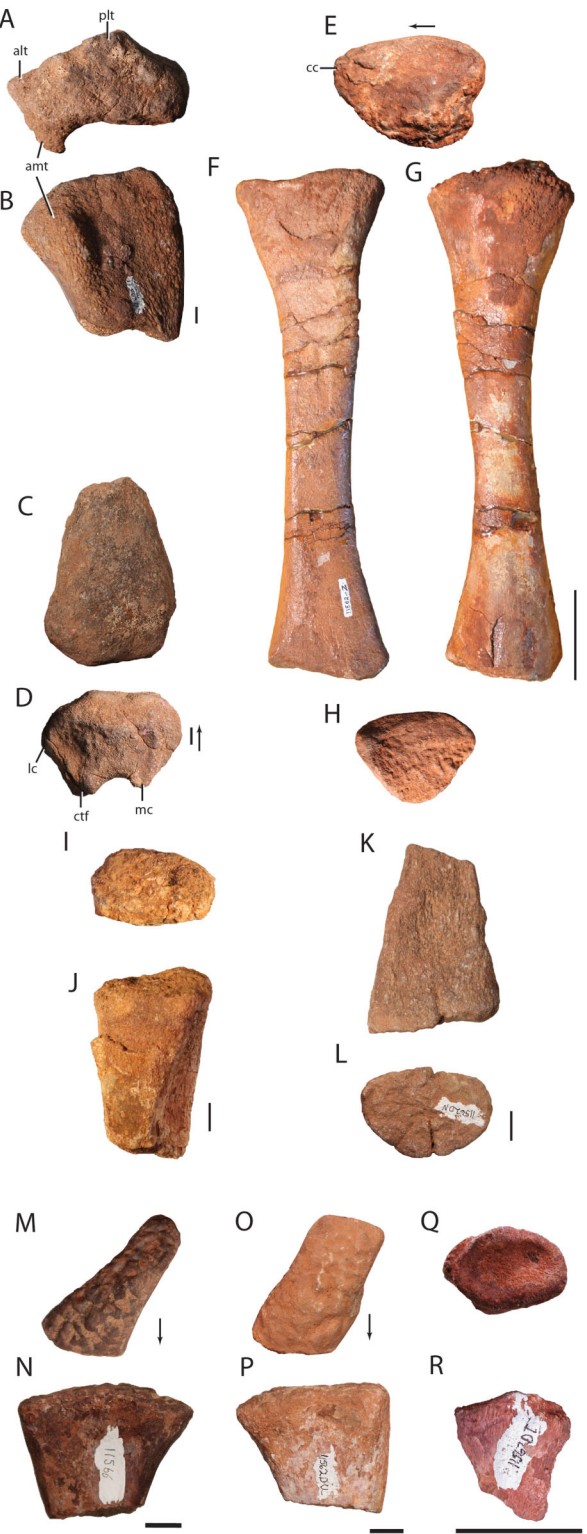

**Figure 13 Hindlimb elements of *Heptasuchus clarki*.** Proximal portion of a right femur (UW 11563-B) in proximal (A) and anterolateral (B) views and the distal portion of the right femur (UW 11563-A) in anterior (C) and distal (D) views; left tibia (UW 11562-Z) in proximal (E), posterior (F), anterior (G), and distal (H) views; proximal portion of a right fibula (UW 11566-S) in proximal (I) and anterolateral

**Figure 13 (continued)**
(J) views and the distal portion of the right fibula (UW 11566-R) in anterior (K) and distal (L) views. Right proximal portion of metatarsal IV (UW 11566) in proximal (M) and ventrolateral (N) views. Possible right metatarsal II (UW 11562DU) in proximal (O) and dorsomedial (P) views. Possible pedal ungual (UW 11562DT) in proximal (Q) and dorsal (R) views. Abbreviations: alc, anterolateral tuber; amt, anteromedial tuber; cc, cnemial crest; ctf, crista tibiofibularis; lc, lateral condyle; mc, medial condyle; plt, posterolateral tuber. Arrows indicate anterior direction. Scales = 1 cm in (A–D), I–R and = 5 cm in (E–H).

tapers to an anteroposteriorly thinner apron medially. The lateral surface of the shaft is smooth without any ridges.

The distal end expands in the last tenth of the length of the pubis. In lateral view, the anterior end slightly expands at its distalmost margin whereas the posterior edge expands comparatively more to form an asymmetric expansion (or boot). The distal margin, in lateral view, is rounded. In anterior view, the pubis shaft medial to the distal expansion is directed posteromedially where it presumably meets its antimere. Consequently, the posteromedial surface of the pubis is distinctly concave in distal view (Fig. 12D). The configuration is in contrast to that of *Batrachotomus kupferzellensis* (SMNS 80279), *Arizonasaurus babbitti* (MSM 4590), *Postosuchus alisonae* (NCSM 13731), and *Poposaurus gracilis* (TMM 43683-1), where the apron is orientated directly medially (*Nesbitt, 2011*). The shape of the distal expansion of *Heptasuchus clarki* is rounded like that of *Batrachotomus kupferzellensis* (SMNS 80279) but not the mediolaterally narrower expansions of poposauroids (*Nesbitt, 2011*). The distal surface is rugose.

**Ischium:** The proximal portion of the right ischium (UW 11564-B; Fig. 12E) was recovered. The proximal portion of the ischium bears a well-defined ridge that demarcates the posteroventral portion of the acetabulum, as in *Batrachotomus kupferzellensis* (SMNS 52970). The robust proximal portion has two articulation surfaces at its proximal edge, a dorsal one for articulation with the ilium and a ventral one for articulation with the pubis. The dorsal and ventral articular surfaces are divided in lateral view by a portion of the ischium that may not have articulated with either the ilium or the pubis. Therefore, there may have been a slight gap between the ischium, ilium, and pubis, like that reconstructed for *Batrachotomus kupferzellensis* (*Gower & Schoch, 2009*; Fig. 6E). Just posterior to the acetabular rim, a clear pit is present on the dorsal edge. This pit occurs in a variety of archosauromorphs (*Ezcurra, 2016*) although its length and form differ among archosaurs (*Gower & Schoch, 2009*).

The shape of the shaft cannot be determined with the preserved portion. The medial surface of the proximal portion of the ischium is flat and the medial and ventral edges indicate that the ischia contacted each other near the proximal portion, similar to other paracrocodylomorphs (e.g., *Postosuchus kirkpatricki*, *Weinbaum, 2013*; *Batrachotomus kupferzellensis*, SMNS 52970; *Arizonasaurus babbitti*, *Nesbitt, 2005*).

**Femur:** Two badly worn fragments representing the proximal and distal ends of a right femur (UW 11563-B, UW 11563-A, respectively; Figs. 13A–13D) were recovered; it is not clear if both ends belong to the same bone. The proximal surface bears a groove like

that of poposauroids and some loricatans (e.g., *Postosuchus kirkpatricki*, *Weinbaum, 2013*) and all three proximal tubera (sensu *Nesbitt, 2005*; *2011*) appear to be present, although the anteromedial tuber is highly eroded (Figs. 13A–13D). The preserved portions of the shaft appear to be thin walled like other paracrocodylomorphs (*Nesbitt, 2011*), but the exact ratio of the thickness of the cortex versus the diameter could not be determined. The distal end bears a small crista tibiofibularis crest and a clear depression is located on the distal surface.

**Tibia:** The well preserved and complete left tibia of *Heptasuchus clarki* (UW 11562-Z; Figs. 13E–13H) is robust with a wide midshaft compared to the length (= 24.0 cm) of the element. The proximal portion does not expand as much relative to the shaft like in *Batrachotomus kupferzellensis* (SMNS 52970), where the proximal portion is more greatly expanded. The proximal surface (maximum length = 7 cm) is roughly triangular with a short cnemial crest and rounded lateral surface for contact with the fibula. The lateral portion of the proximal surface is depressed like that of suchian archosaurs (*Nesbitt, 2011*) and this surface is separated from the posterior portion of the tibia by a vertical gap (Fig. 13). The proximal surface is highly rugose.

The shaft of the tibia remains oval in section throughout its length, and like the femur, the tibia is also thin walled. The posterior surface of the entire bone, in contrast to the other faces, is flattened, and exhibits a slight twisting along its length. The distal end of the tibia (maximum width = 6 cm) is expanded less than the proximal end and is triangular in distal view. The differentiation of the distal surface of the tibia for articulation with the astragalus is poor; the 'cork-screw' configuration (proximally slanted posterolateral surface and distally expanded anteromedial portion) typical in shuvosaurids (*Nesbitt, 2007*), aetosaurs (*Parrish, 1993*), *Batrachotomus kupferzellensis* (SMNS 52970) and in rauisuchid taxa like *Postosuchus kirkpatricki* (TTUP 9002) is not present in *Heptasuchus clarki*. Instead the distal surface is flatter in *Heptasuchus clarki* and is more like that of *Prestosuchus chiniquensis* (*Huene, 1942*; *Desojo, Baczko & Rauhut, 2020*). The distal surface is also rugose.

**Fibula**: The fibula is only represented by the right (?) proximal portion (UW 11566-S) and right distal portion (11566-R) recovered among weathered fragments (Figs. 13I–13L). The more robust proximal portion is asymmetrical in lateral view with a tapering posterior portion. The distal end expands anteriorly and posteriorly and possesses an ovate distal surface (with an anteroposteriorly long axis).

**Metatarsals and phalanges:** A number of fragmentary metatarsals (UW 11562, UW 11562-DH, UW 11562-DHU, UW 11562-DR) and phalanges were recovered from the locality and all pes elements consist of weathered proximal or distal ends. Given the difficulty of assigning fragments of metatarsal, we are hesitant to assign anatomical positions to the most fragments, but we have identified a proximal end of a right metatarsal IV (UW 11566; Figs. 13M and 13N) and possibly a proximal end of a right metatarsal II (UW 11566; Figs. 13O and 13P) based on comparisons with the pes of *Postosuchus alisonae* (NCSM 13731). The proximal surfaces of the metatarsals have rugose surfaces and

are typically rectangular with well-defined faces with squared-off ventral ends of the proximal surfaces. The distal end of the metatarsals poses large articular facets that are about as long as wide. A single ungual (UW 11562-DT; Figs. 13O and 13P), possibly from the pes, indicates that the unguals were dorsoventrally flattened like that of *Prestosuchus chiniquensis* (*Huene, 1942*).

## Phylogenetic Analysis

The phylogenetic position of *Heptasuchus clarki* was assessed using the early archosaur matrix of *Nesbitt (2011)* as a base followed by the modifications of characters, scores, and terminal taxa of *Butler et al. (2014, 2018)*, *Nesbitt et al. (2014, 2017, 2018)*, *Nesbitt & Desojo (2017)* and *Desojo, Baczko & Rauhut (2020)* and additions of terminal taxa by *von Baczko, Desojo & Pol (2014)*, *Lacerda et al. (2016)* and *Lacerda, De França & Schultz (2018)*. We added the additional and new characters of *Desojo, Baczko & Rauhut (2020*; characters 414–422 here), the aphanosaur-centered characters of *Nesbitt et al. (2017*; characters 434–439 here), a character for rauisuchids and kin from *Brusatte et al. (2008*; *2010*; character 424 here), and nine new characters centered on the relationships of *Heptasuchus clarki* among loricatans (Fig. 14; characters 425–433 here; see appendix 1) for a total of 439 characters. Characters 32, 52, 121, 137, 139, 156, 168, 188, 223, 247, 258, 269, 271, 291, 297, 314 328, 356, 371, 399 and 413 were ordered—21 total. We ordered characters 314 and 371 based on the character descriptions of *Nesbitt (2011)*—characters were not listed in the ordered state list in character sampling and methods. The characters were scored in Mesquite (*Maddison & Maddison, 2015*).

Our primary dataset consists of 100 terminal taxa (Supplemental Information). This dataset now contains the most specimens and species level terminal taxa of paracrocodylomorphs to date. The matrix includes some stem archosaurs, but for better taxon and character sampling of this part of the tree see *Ezcurra (2016)* and likewise, for better taxon and character sampling for Dinosauria see the dataset of *Baron, Norman & Barrett (2017a)* and further modifications (*Langer et al., 2017*; *Baron, Norman & Barrett, 2017b*).

The matrix was constructed in Mesquite (Madison & Madison, 2015) and analyzed with equally weighted parsimony using TNT v. 1.5 (*Goloboff & Catalano, 2016*). Using parsimony, we used new technology search (with the following boxes checked: Sectorial Search, Drift, and Tree Fusing) until 100 hits to the same minimum length. These trees were then run through a traditional search (search trees from RAM option) using TBR branch swapping. *Euparkeria* was set as the outgroup. Zero length branches were collapsed if they lacked support under any of the most parsimonious reconstructions.

We ran the first analysis a priori excluding the following terminal taxa: *Lewisuchus admixtus*, *Pseudolagosuchus majori* (combined into *Lewisuchus/Pseudolagosuchus* following *Nesbitt et al., 2010*, *Nesbitt, 2011* and *Ezcurra et al., 2019*), '*Prestosuchus loricatus* paralectotype' (*Desojo, Baczko & Rauhut, 2020*), and collapsed *Prestosuchus chiniquensis* lectotype, *Prestosuchus chiniquensis* paralectotype, *Prestosuchus chiniquensis* type series, UFRGS PV 156 T, UFRGS PV 152 T, CPEZ 239b into a '*Prestosuchus chiniquensis* ALL' (with the addition of scores from ULBRA-PVT-281; Roberto-Da-Silva et al. 2018), added

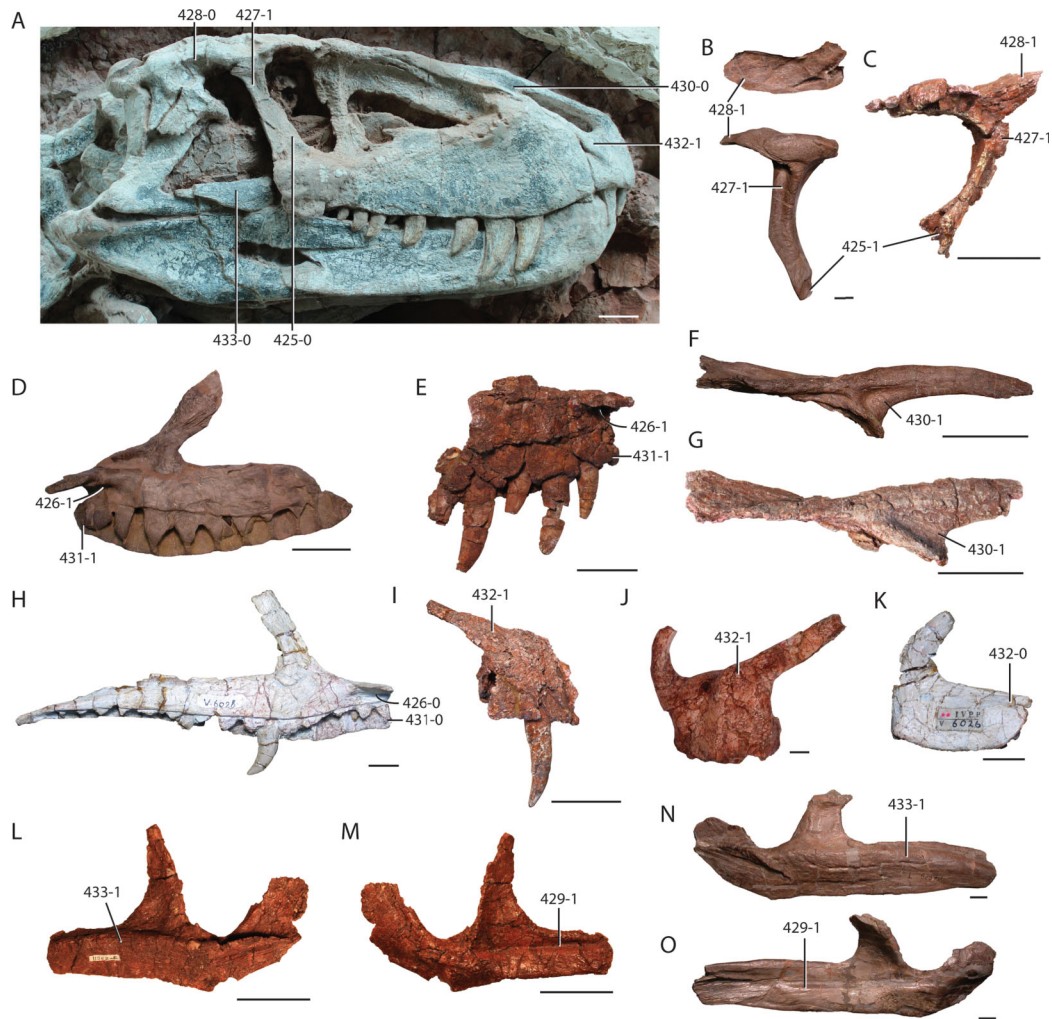

**Figure 14 New illustrated character states for paracrocodylomorph archosaurs.** (A) skull referred to *Prestosuchus chiniquensis* (ULBRA-PVT-281) in right lateral view; (B) right postorbital of *Batrachotomus kupferzellensis* (SMNS 52970) in dorsal (top) and lateral (bottom) view; (C) left postorbital of *Heptasuchus clarki* (UW 11562) in lateral view; (D) left maxilla of *Batrachotomus kupferzellensis* (SMNS 52970) in medial view; (E) right maxilla of *Heptasuchus clarki* (UW 11562) in medial view; (F) right nasal of *Batrachotomus kupferzellensis* (SMNS 52970) in lateral view; (G) right nasal of *Heptasuchus clarki* (UW 11562) in lateral view; (H) left maxilla of *Xilousuchus sapingensis* (IVPP V6026) in medial view; (I) right premaxilla of *Heptasuchus clarki* (UW 11562) in lateral view; (J) left premaxilla of *Postosuchus kirkpatricki* (TTUP 9000) in lateral view; (K) left premaxilla of *Xilousuchus sapingensis* (IVPP V6026) in lateral view; (L) right jugal of *Heptasuchus clarki* (UW 11562) in lateral view; (M) right jugal of *Heptasuchus clarki* (UW 11562) in medial view; (N) left jugal of *Batrachotomus kupferzellensis* (SMNS 52970) in lateral view; (O) left jugal of *Batrachotomus kupferzellensis* (SMNS 52970) in medial view. Numbers refer to character number separated by a dash from the state. Scales in 10 cm in A, 5 cm in (C–G), (I), (L–M), and 1 cm in (B), (H), (J–K), (N–O).

to another description (UFRGS-PV-0629-T; *Mastrantonio et al., 2019*; see Supplemental Information). This data matrix resulted in 144 most parsimonious trees (MPTs) of length (1553 steps) (Consistency Index = 0.330; Retention Index = 0.749) (See Supplemental Information for full tree; S1).

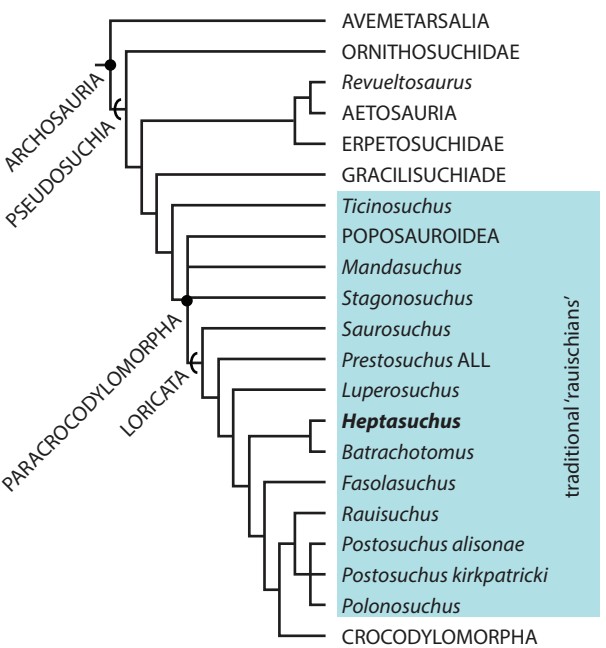

**Figure 15 Partial phylogenetic tree focused on pseudosuchian relationships with *Heptasuchus clarki* included.** *Heptasuchus clarki* was found as a loricatan as the sister-taxon of *Batrachotomus kupferzellensis*. Tree derived from 72 most parsimonious trees (MPTs) of length (1529 steps) (Consistency Index = 0.335; Retention Index = 0.752) (see Figures S1 and S2 for the full phylogenetic tree).

In our main analysis, we also eliminated *Nundasuchus songeaensis* and *Pagosvenator candelariensis* from the final analysis because (1) *Nundasuchus songeaensis* likely is closer to the base of Archosauria (see *Nesbitt et al., 2014*) and (2) *Pagosvenator candelariensis* is clearly a member of Erpetosuchidae (*Lacerda, De França & Schultz, 2018*), but because of missing information and some character conflict, the taxon is highly unstable (see *Desojo, Baczko & Rauhut, 2020*). Both taxa could thus greatly impact the optimizations of character states at the base of and within Paracrocodylomorpha which is the target portion of the Pseudosuchian tree here. This data matrix resulted in 72 most parsimonious trees (MPTs) of length 1529 steps (Consistency Index = 0.335; Retention Index = 0.752) (Fig. 15 for partial tree; See Supplemental Information for full tree; S2).

# DISCUSSION

## The phylogenetic position of *Heptasuchus clarki* among archosaurs

The results of both our analyses (Supplemental Information) is similar to the original analysis of *Nesbitt (2011)* where classic "Rauisuchia" is a paraphyletic group relative to Crocodylomorpha with "Rauisuchia" divided among loricatans (paracrocodylomorph taxa closer to Crocodylomorpha), poposauroids (paracrocodylomorph taxa closer to *Shuvosaurus inexpectatus*), and a few taxa just outside Paracrocodylomorpha (e.g., *Mandasuchus tanyauchen*, *Ticinosuchus ferox*) (Fig. 15). Unsurprisingly, this pattern has been retained in most iterations of the *Nesbitt (2011)* dataset (*Butler et al., 2011*, *2014*, *2018*; *von Baczko, Desojo & Pol, 2014*; *Lacerda et al., 2016*; *Lacerda, De França & Schultz, 2018*;

*Nesbitt & Desojo, 2017*; *Nesbitt et al., 2014*, *2017*, *2018*; *Desojo, Baczko & Rauhut, 2020*).
Like these other analyses, the base of Paracrocodylomorpha is poorly supported with the
addition or removal of a taxon, a character score change, or the addition of new characters
that alter the relationships of early diverging taxa such as *Mandasuchus tanyauchen* and
*Stagonosuchus nyassicus*. Within Loricata, *Saurosuchus galilei*, *Prestosuchus chiniquensis*, and
*Luperosuchus fractus* consistently are located at the base of the clade. The relationship of
these taxa could be a grade (as found here) or in a clade (*Nesbitt & Desojo, 2017*; *Desojo,
Baczko & Rauhut, 2020*) as a consequence of character optimizations for taxa closer to
Crocodylomorpha. Moreover, we did not find *Stagonosuchus* (=*Prestosuchus* *Desojo,
Baczko & Rauhut, 2020*) *nyassicus* as the sister taxon of *Prestosuchus chiniquensis* with the
addition of our new characters (see appendix), but given that the new characters focus on the
skull and *Stagonosuchus nyassicus* is almost entirely represented by postcrania, this
instability is not surprising. The relationship within loricatans closer to Crocodylomorpha
(e.g., *Batrachotomus kupferzellensis* + *Alligator mississippiensis*) remained unchanged in
comparison with *Nesbitt (2011)*.

*Heptasuchus clarki* is well nested within Loricata and firmly supported as the sister
taxon of *Batrachotomus kupferzellensis*. The following four unambiguous character states
support the sister taxon relationship within Loricata where *Heptasuchus clarki* could be
scored: posterior portion of the nasal is concave at the midline in dorsal view (34-1);
supratemporal fossa present anterior to the supratemporal fenestra (144-1); ventral surface
of palatal process of the maxilla with distinct fossa (426-1); medial side of the posterior
process of the jugal with longitudinal groove (429-1). The following nine unambiguous
character states are synapomorphies within Loricata and scored for *Batrachotomus
kupferzellensis*, but not for *Heptasuchus clarki* because of missing information: dorsal
(=ascending) process of the maxilla remains the same width for its length (29-0); anterior
portion of the frontal tapers anteriorly along the mid-line (43-1); squamosal with
distinct ridge on dorsal surface along edge of supratemporal fossa (49-1); upper temporal
fenestrae of the parietal by a mediolaterally thin strip of flat bone separated (59-1);
double-headed ectopterygoid (89-1) (the jugal indicates that the ectopterygoid was likely
double-headed, but we chose not to score it because the ectopterygoid was not preserved);
articular with dorsomedial projection separated from glenoid fossa by a clear concave
surface (156-1); angle between the lateral condyle and the crista tibiofibularis of the femur
about a right angle in distal view (319-1); presacral and paramedian osteoderms with a
distinct longitudinal bend near lateral edge (404-1); presacral osteoderms dimensions
longer than wide (407-1); position of the posterior process of the squamosal below anterior
process and set off by distinct step (423-1).

*Heptasuchus clarki* is well supported as the sister taxon of *Batrachotomus
kupferzellensis*. The following unambiguous character states support this relationship:
anterodorsal margin at the base of the dorsal process of the maxilla concave (25-1);
dorsolateral margin of the anterior portion of the nasal with a distinct anteroposteriorly
ridge on the lateral edge (35-1); depression on the anterolateral surface of the ventral end
of the postorbital (425-1)(also present in *Postosuchus kirkpatricki*); distinct fossa on the
posterodorsal portion of the naris on the lateral side of the nasal (430-1); anteroposteriorly

trending ridge on the lateral side of the jugal is asymmetrical dorsoventrally where the dorsal portion is more laterally expanded (433-1). The crania of *Heptasuchus clarki* share a number of unique features with *Batrachotomus kupferzellensis*, many of which were once considered autapomorphies of *Batrachotomus kupferzellensis* (*Gower, 1999*). However, we were not able to pinpoint any postcranial character states that *Batrachotomus kupferzellensis* and *Heptasuchus clarki* share exclusively.

### *Heptasuchus clarki* and *Poposaurus gracilis*

When initially described, *Heptasuchus clarki* was considered to be from the Popo Agie Formation, which also contained the remains of another 'rauisuchian' *Poposaurus gracilis*. *Long & Murry (1995)* hypothesized that *Heptasuchus clarki* may be a poposauroid after comparisons with *Poposaurus gracilis*, *Shuvosaurus* (= 'Chatterjeea') *elegans* and *Postosuchus kirkpatricki*. Soon after, *Zawiskie & Dawley (2003)* hypothesized that the skull of *Heptasuchus clarki* might belong to the body of *Poposaurus gracilis* based on age proximity and on a few overlapping postcranial bones. After further analyses, we now reject these hypotheses based on a number of lines of evidence. First of all, our robust phylogenetic analysis clearly places *Heptasuchus clarki* and *Batrachotomus kupferzellensis* as close relatives and both are more closely related to crocodylomorphs than poposauroids. Second, the deposits that *Heptasuchus clarki* was found in are likely not the same as the Popo Agie Formation from the western portion of Wyoming and the deposits that *Heptasuchus clarki* was found in are likely older than that of the Popo Agie Formation and hence *Poposaurus gracilis*. Third, with an abundance of new specimens of *Poposaurus gracilis* from partial skeletons (*Weinbaum & Hungerbühler, 2007*) to nearly complete and articulated postcranial remains (*Gauthier et al., 2011*; *Schachner et al., 2019*), and comparative skull material (*Parker & Nesbitt, 2013*), it is clear that *Poposaurus gracilis* and *Heptasuchus clarki* are different taxa.

### Further implications of *Heptasuchus clarki*

The stratigraphic and temporal occurrence of *Heptasuchus clarki* fills a critical gap in loricatan biogeography within current-day North America and across Pangea. *Heptasuchus clarki* is the only confirmed loricatan taxon from either the late Middle Triassic or the early portion of the Late Triassic (see above) and demonstrates that large paracrocodylomorphs were present from the early portion of the Middle Triassic (i.e., *Arizonasaurus babbitti* and other forms from the Moenkopi Formation; *Nesbitt, 2003*; *Schoch et al., 2010*) through the end of the deposition of Upper Triassic strata (*Effigia okeeffeae* "siltstone member", *Coelophysis* Quarry, *Redondavenator quayensis Nesbitt et al., 2005*). Furthermore, *Heptasuchus clarki* fills a 'phylogenetic gap' in that it is the only named loricatan from current-day North America that does not fit into Poposauroidea (Ctenosauriscidae or Shuvosauridae), Rauisuchidae (e.g., *Postosuchus*, *Viviron haydeni*), or Crocodylomorpha and links these disparate clades present in current-day North America to forms from current-day South America and Europe. The presence of a 'mid-grade' loricatan in current-day North America hints that earlier diverging loricatans known from current-day South America (*Prestosuchus chiniquensis*, *Luperosuchus fractus*, *Saurosuchus*

*galilei*) may have had close relatives in current-day North America, but equivalently-aged deposits in North America are lacking.

The sister taxon relationship of *Heptasuchus clarki* and *Batrachotomus kupferzellensis* demonstrates the first biotic link between current-day North America in the Middle to early Late Triassic and the Middle Triassic (Ladinian Stage) of current-day Germany. Although the assemblage from the *Heptasuchus clarki* bonebed has not been studied in detail (see above), there are no other overlapping species or genus-level taxa that are present from the *Heptasuchus clarki* bonebed and the *Batrachotomus kupferzellensis* locality (= Kupferzell = Lagerstätte Kupferzell-Bauersbach), let alone major clades (e.g., the temnospondyls *Gerrothorax*, *Plagiosuchus*, *Mastodontosaurus*, *Kupferzellia*, *Trematolestes*, the chronosuchian *Bystrowiella schumanni*, Choristodera, the sauropterygian *Nothosaurus*; *Hagdorn et al., 2015* and a variety of smaller tetrapods represented by jaw material or tooth distinct morphologies; *Schoch et al., 2018*). Moreover, the clades present in the Ladinian-aged Kupferzell locality of current-day Germany are either completely absent or rare in North America during the entire Triassic Period (e.g., the temnospondyl clades from the Lagerstätte Kupferzell-Bauersbach, chronosuchian). The similarity of just the large carnivorous archosaurs between current day North America and Germany in highly differentiated vertebrate assemblages implies that the larger archosaurs may have had significant flexibility in their paleoenvironments across Pangea through the Middle to Upper Triassic. This notion is further supported by the evidence presented by *Nesbitt et al. (2009)* suggesting that carnivorous archosaurs (e.g., dinosaurs and crocodylomorphs) may have had greater distribution in the environments across Pangea.

The holotype locality of *Heptasuchus clarki* contains a minimum of four individuals and this occurrence appears to be common with paracrocodylomorph archosaurs, at least in the Triassic Period. The exact number of individuals is not known because of the heavily weathered bonebed, but it is clear that some individuals were highly scattered and disarticulated whereas some other individuals, including the holotype, were closely associated. The closest relative of *Heptasuchus clarki*, *Batrachotomus kupferzellensis* was also found in a similar condition: associated and disarticulated individuals across a bonebed (i.e., Lagerstätte Kupferzell-Bauersbach; *Gower, 1999*). Finding non-crocodylomorph paracrocodylomorphs (or "rauisuchians") in bonebeds with more than one individual appears common across the clade from the Middle to Upper Triassic across Pangea. For example, multiple individuals of *Heptasuchus clarki*, *Batrachotomus kupferzellensis*, *Postosuchus kirkpatricki*, *Effigia okeeffeae*, *Shuvosaurus inexpectatus*, and *Decuriasuchus quartacolonia* have been found together in the same deposits. The preservation of these paracrocodylomorphs ranges from nearly complete skeletons to disarticulated, but associated skeletons. The implications of the association of these individuals to behavior must be carefully considered on a variety of anatomical, taphonomic and sedimentalogical data (*De França, Ferigolo & Langer, 2011*), but the repeated co-occurrence of individuals of paracrocodylomorphs is intriguing and may suggest that these reptiles were typically in groups (*De França, Ferigolo & Langer, 2011*) and that this behavior was maintained through much of their evolutionary history.

## APPENDIX

**New character descriptions (see Fig. 14):**

425. Postorbital, ventral end, depression on the anterolateral surface: (0) - absent; (1) - present. (new; Fig. 14)

The plesiomorphic condition, state 0, in stem archosaurs and within Archosauria is to have a tapering ventral end of the postorbital that fits onto the anterodorsal edge of the dorsal process of the jugal and this condition is clear in the following exemplary taxa: *Euparkeria capensis* (*Ewer, 1965*); *Lewisuchus admixtus* (*Bittencourt et al., 2015*); *Gracilisuchus stipanicicorum* (MCZ 4117), *Paratypothorax andressorum* (SMNS 19003; *Schoch & Desojo, 2016*) and *Luperosuchus fractus* (PULR 04). In a number of loricatan taxa (e.g., *Batrachotomus kupferzellensis*, SMNS 80260; *Heptasuchus clarki*, UW 11562; and to a lesser degree *Postosuchus kirkpatricki*, TTUP 9000), the ventral end of the postorbital extends anteriorly into the orbit (*Benton & Clark, 1988*; *Juul, 1994*; *Benton, 1999*; *Alcober, 2000*; *Benton & Walker, 2002*; *Brusatte et al., 2010*; *Nesbitt, 2011* Character 65). Out of these taxa, the ventral end of the postorbital is flat or nearly flat whereas a depression on the ventrolateral portion of the distal end of the postorbital is present in both *Batrachotomus kupferzellensis* (SMNS 80260), *Postosuchus kirkpatricki* TTUP 9000, and *Heptasuchus clarki* (UW 11562) – state 1. *Gower (1999)* listed the depression as a possible autapomorphy of *Batrachotomus kupferzellensis*. The ventrolateral depression in *Heptasuchus clarki* is much deeper and much of the depth is hidden in lateral view compared to *Batrachotomus kupferzellensis* and *Postosuchus kirkpatricki*.

426. Maxilla, medial side, ventral surface of palatal process: (0) flat; (1) - depression present. (new; Fig. 14)

The palatal process of the maxilla is horizontal in most archosauriforms and the ventral surface of the palatal process is typically flat or slightly concave. Within Pseudosuchia, the ventral surface of the palatal process is flat in *Xilousuchus sapingensis* (*Nesbitt, Liu & Li, 2011*), *Revueltosaurus callenderi* (PEFO 34561) and in the ornithosuchid *Riojasuchus tenuisceps* (PVL 3827; *von Baczko & Desojo, 2016*). In contrast, a dorsally extended depression at the posteroventral side of the palatal process of the maxilla is present in *Postosuchus kirkpatricki* (TTUP 9000), *Polonosuchus silesiacus* (ZPAL Ab III/543), *Fasolasuchus tenax* (PVL 3851), *Heptasuchus clarki* (UW 11562). *Batrachotomus kupferzellensis* (SMNS 80260), *Arganosuchus dutuit* (ALM 1; *Jalil & Peyer, 2007*) and possibly in *Sphenosuchus actus* (SAM 3014). It appears that the depression is not present in any of the *Prestosuchus chiniquensis* specimens where the palatal process is visible (*Mastrantonio et al., 2019*). In some taxa (e.g., *Postosuchus kirkpatricki*, TTUP 9000) the depression is much deeper in that the depression extends well dorsal to the dorsal extent of the palatal process whereas in *Sphenosuchus actus*, the depression is rather shallow but occurs in the same position as that of other loricatans. The function of the depression is not clear. *Chatterjee (1985)* hypothesized that the depression could serve as the area for Jacobson's organ. However, *Weinbaum (2011)* points out that Jacobson's organ is not present in crocodylians and avians and thus unlikely that this depression was for housing

Jacobson's organ. The depression is located too far medially and, in most taxa, dorsally to represent a depression for accepting an enlarged dentary tooth.

427. Postorbital, lateral side, posterodorsal portion of the ventral process: (0) – smooth; (1) – slight depression, usually ventral to a rounded knob or ridge. (new; Fig. 14)

The posterior side of the postorbital is typically bowed or flat similar to the anterior and lateral sides of the base of the ventral process. Examples of taxa with this plesiomorphic condition include *Euparkeria capensis* (*Ewer, 1965*); *Lewisuchus admixtus* (Bittencourt et al. 2014); *Gracilisuchus stipanicicorum* (MCZ 4117), and *Paratypothorax andressorum* (SMNS 19003; *Schoch & Desojo, 2016*). Within Paracrocodylomorpha, *Luperosuchus fractus* (PULR 04), and *Xilousuchus sapingensis* (Nesbitt et al. 2011) have state 0. In *Prestosuchus chiniquensis* (UFRGS-PV-0629-T; *Mastrantonio et al., 2019*), *Postosuchus kirkpatricki* (TTUP 9000), *Batrachotomus kupferzellensis* (SMNS 80260), *Heptasuchus clarki* (UW 11562), *Arizonasaurus babbitti* (MSM 4590), and *Sphenosuchus actus* (SAM 3014) have a clear depression on the posterior side of the ventral process of the postorbital near its base (i.e., near the contact with the squamosal. The taxa scored as state 1 typically have a vertical ridge, sometimes rugose, that divide the anterior part of the ventral process of the postorbital from the posterior portion.

428. Squamosal - postorbital articulation: (0) - postorbital fits into a groove on the lateral side of the squamosal; (1) - the postorbital lies on the dorsal surface of the squamosal; (2) - the squamosal largely lies on the dorsal surface of the postorbital. (new; Fig. 14)

In stem archosaurs and most members of Archosauria, the posterior portion of the postorbital fits into a clear slot into the lateral side of the squamosal. Clear examples of this articulation include *Euparkeria capensis* (*Ewer, 1965*), *Arizonasaurus babbitti* (MSM 4590), *Paratypothorax andressorum* (SMNS 19003; *Schoch & Desojo, 2016*), and *Riojasuchus tenuisceps* (PVL 3827; *von Baczko & Desojo, 2016*). In most loricatans, the anterior process of the squamosal largely fits on the dorsal surface of the postorbital (state 1). As noted by *Gower (1999)* for *Batrachotomus kupferzellensis*, much of the squamosal of the taxon dorsally overlaps the postorbital, but there is some complexity to this articulation; a small part of the posteromedial portion of the postorbital is underlapped by the squamosal, and this results in the postorbital lying in a small notch of the squamosal. Early diverging loricatans *Luperosuchus fractus* (*Nesbitt & Desojo, 2017*), *Prestosuchus chiniquensis* (UFRGS-PV-0629-T), and *Saurosuchus galilei* (PVSJ 32) appear to have state 1, although it is a bit difficult to see the articulation in the specimens represented by partially articulated or fully articulated skulls. State 1 is clearly present in *Batrachotomus kupferzellensis* (SMNS 80260), *Heptasuchus clarki* (UW 11562), and *Postosuchus kirkpatricki* (TTUP 9000). Within Crocodylomorpha, state 2 appears to be present across the clade where the postorbital largely lies over the squamosal and this is clear in early members of crocodylomorphs like *Dromicosuchus grallator* (NCSM 13733), *Dibothrosuchus elaphros* (IVPP V 7907), and *Litargosuchus leptorhynchus* (*Clark & Sues, 2002*). Crocodyliforms appear to have an interdigitating suture between the postorbital and squamosal so these taxa are scored as ?
429. Jugal, posterior process, medial side, longitudinal groove: (0) – absent; (1) - present. (new; Fig. 14)

Typically, the medial surface of the posterior process of the jugal of stem archosaurs (e.g., *Euparkeria capensis*) and members of Archosauria (e.g., *Arizonasaurus babbitti*, MSM 4590; *Effigia okeeffeae*; *Nesbitt, 2007*) are smooth. A clear groove, that parallels the ventral edge is present for nearly the entire length of the jugal in *Batrachotomus kupferzellensis* (SMNS 52970), *Postosuchus kirkpatricki* (TTUP 9000), *Polonosuchus silesiacus* (ZPAL Ab III/543), *Heptasuchus clarki* (UW 11562), and *Sphenosuchus actus* (SAM 3014).

430. Nasal, posterodorsal corner of the naris: (0) - smooth or slight fossa; (1) - distinct fossa with a rim present. (new; Fig. 14)

The anterior portion of the nasal of archosaurs typically splits into a process that lies dorsal to the external naris and one that extends anteroventrally posterior of the external naris (=descending process of some). In the juncture of the two anterior processes, the surface is typically flat. This is the case in most loricatans (e.g., *Postosuchus kirkpatricki*; TTUP 9000; specimens referred to *Prestosuchus chiniquensis*). In *Batrachotomus kupferzellensis* (SMNS 52970) and *Heptasuchus clarki* (UW 11562), there is a clear narial fossa (sensu *Gower, 1999*) between the two anterior processes. Ventral to this fossa, a ridge framing the fossa is present on the anteroventral process in these taxa. This depression is not the fully the consequence of the ridge present dorsally (character 35, state 1) given that *Postosuchus kirkpatricki* (TTUP 9000) possesses that ridge, but not the fossa. A Moenkopi form (NMMNH 55779; *Schoch et al., 2010*) also possesses state 1.

431. Maxilla, anteroventral corner: (0) - abuts premaxilla; (1) - extensively laterally overlaps the posteroventral corner of the premaxilla. (new; Fig. 14)

Within stem archosaurs and within Archosauria, the juncture between the maxilla and premaxilla at their ventral borders is either separated by a gap (e.g., *Riojasuchus tenuisceps*, *von Baczko & Desojo, 2016*; *Coelophysis bauri*, *Colbert, 1989*), or is loosely connected (e.g., *Euparkeria capensis*, *Ewer, 1965*; *Turfanosuchus dabanensis*, IVPP V3237). In loricatans, there is a medially extended articulation surface between the maxilla and premaxilla. Here, the anterolateral portion of the maxilla lies onto a clear articulation surface on the posterolateral side of the premaxilla. This character state (1) is present in *Saurosuchus galilei* (PVSJ 32), *Batrachotomus kupferzellensis* (SMNS 52970), *Heptasuchus clarki* (UW 11562), *Polonosuchus silesiacus* (ZPAL Ab III/543), *Postosuchus kirkpatricki* (TTUP 9000), and *Fasolasuchus tenax* (PVL 3851). The state in crocodylomorphs is not clear.

This character is difficult to score in articulated skulls because the targeted surfaces cannot be seen so we recommend only scoring the character if the maxilla and premaxilla are disarticulated and the anterior end of the maxilla is complete. Fine surface preservation is typically required also. Additionally, it is possible that this character is correlated with larger sizes; that is, it is easier to see in larger specimens.

432. Premaxilla, base of the posterodorsal process (maxillary process): (0) - flat with the body of the premaxilla; (1) - laterally bulging from the main body. (new; Fig. 14)

The base of the posterodorsal process of the premaxilla is typically continuous with the lateral surface of the body of the premaxilla in stem archosaurs (e.g., *Euparkeria capensis*, *Ewer, 1965*; *Erythrosuchus africanus*, BPI 4526). Within Archosauria, state 0 is typical of avemetatarsalians (e.g., *Silesaurus opolensis*; Dzik 2003; *Coelophysis bauri*, *Colbert, 1989*) and occurs throughout early diverging Pseudosuchia (e.g., *Xilousuchus sapingensis*, IVPP V6026; *Paratypothorax andressorum*, SMNS 19003; *Riojasuchus tenuisceps*, PVL 3827). In Loricata, *Prestosuchus chiniquensis* (ULBRA-PVT-281), *Saurosuchus galilei* (PVSJ 32), *Heptasuchus clarki* (UW 11562), *Postosuchus kirkpatricki* (TTUP 9000), *Fasolasuchus tenax* (PVL 3850), and *Polonosuchus silesiacus* (ZPAL Ab III/543) all have laterally expanded base of the posterodorsal process of the premaxilla. The bulge is much clearer in some taxa (e.g., *Postosuchus kirkpatricki* TTUP 9000) than others (e.g., *Saurosuchus galilei*, PVSJ 32). Early crocodylomorphs (e.g., *Dromicosuchus grallator*, NCSM 13733) appear to also have state 1.

433. Jugal, lateral surface, anteroposteriorly trending ridge: (0) - symmetrical dorsoventrally; (1) - asymmetrical dorsoventrally where the dorsal portion is more laterally expanded. (new; Fig. 14)

The lateral surface of the jugal of archosaurs is either smooth or bears a ridge that parallels the ventral edge (character 75 of *Nesbitt, 2011*). The form of the ridge varies across Archosauria and can be a sharp ridge, broad, or laterally extended as a rugose and broad ridge. Most loricatans have some kind of ridge, but *Heptasuchus clarki* (UW 11562) and *Batrachotomus kupferzellensis* (SMNS 52970) share a clear expanded ridge that is asymmetrical dorsoventrally where the dorsal portion is more laterally expanded.

## INSTITUTIONAL ABBREVIATIONS

**ALM**      Alili n'yifis' locality near the village of Alma. Specimens stored at Museum National d'Histoire Naturelle, Paris, France (MNHN)

**BPI**      Evolutionary Studies Institute (formerly Bernard Price Institute for Palaeontological Research), University of the Witwatersrand, Johannesburg, South Africa

**CPEZ**      Coleção de Paleontologia do Museu Paleontológico Arqueológico Walter Ilha, São Pedro do Sul, Brazil

**GPIT**      Institut und Museum für Geologie und Paläontologie, Universität Tübingen, Germany

**IVPP**      Institute of Vertebrate Paleontology and Paleoanthropology, Beijing, China

**MSM**      Arizona Museum of Natural History, Mesa, Arizona, USA

**NCSM**      North Carolina Museum of Natural Sciences, Raleigh, North Carolina, USA

**NHMUK**      (formerly BMNH), Natural History Museum, London, U.K

**NMMNH**      New Mexico Museum of Natural History and Science, Albuquerque, New Mexico, USA

**NMT**      National Museum of Tanzania, Dar es Salaam, Tanzania

**PEFO**      Petrified Forest National Park, Arizona, USA

**PULR**      Paleontología, Universidad Nacional de La Rioja, La Rioja, Argentina

| | |
|---|---|
| **PVL** | Paleontología de Vertebrados, Instituto "Miguel Lillo", San Miguel de Tucumán, Argentina |
| **PVSJ** | División de Paleontología de Vertebrados del Museo de Ciencias Naturales y Universidad Nacional de San Juan, San Juan, Argentina |
| **SAM** | Iziko South African Museum, Cape Town, South Africa |
| **SMNS** | Staatliches Museum für Naturkunde, Stuttgart, Germany |
| **SNSB-BSPG** | Staatliche Naturwissenschaftliche Sammlungen Bayerns, Bayerische Staatssammlung für Paläontologie und Geologie, Munich, Germany |
| **TMM** | Texas Vertebrate Paleontology Collections, The University of Texas at Austin, Texas, USA |
| **TTU** | Texas Tech University Museum, Lubbock, Texas, USA |
| **UFRGS-PV** | Laboratório de Paleovertebrados, Universidade Federal do Rio Grande do Sul, Porto Alegre, Brazil |
| **ULBRA-PVT** | Paleovertebrate Collection of the Universidade Luterana do Brasil, Canoas, Rio Grande do Sul, Brazil |
| **USNM** | National Museum of Natural History (formerly United States National Museum), Smithsonian Institution, Washington, D.C., USA |
| **UW** | University of Wyoming, Laramie, Wyoming, USA |
| **ZPAL** | Institute of Paleobiology, Polish Academy of Sciences, Warsaw, Poland |

## ACKNOWLEDGEMENTS

We especially thank Michelle Stocker for conversations about Wyoming geology, Triassic rocks of the western US, and Triassic assemblages. We thank the many insightful conversations about 'rauisuchian' relationships and anatomy with David J. Gower and Julia B. Desojo. We thank Brent Breithaupt for help tracking down BLM permit numbers in challenging times. Laura A. Vietti helped locate and curate *Heptasuchus clarki* material from the UW collection and J. Chris Sagebiel curated the TMM *Heptasuchus clarki* material. Reviews by Andrew Heckert, Thomasz Sulej, and Jonathan Weinbaum greatly improved the paper. We thank the Willi Henning Society for free access to TNT software.

### Funding

The authors received no funding for this work.

### Competing Interests

The authors declare that they have no competing interests.

### Author Contributions

- Sterling J. Nesbitt conceived and designed the experiments, performed the experiments, analyzed the data, prepared figures and/or tables, authored or reviewed drafts of the paper, and approved the final draft.

- John M. Zawiskie conceived and designed the experiments, performed the experiments, prepared figures and/or tables, authored or reviewed drafts of the paper, and approved the final draft.
- Robert M. Dawley conceived and designed the experiments, performed the experiments, authored or reviewed drafts of the paper, and approved the final draft.

## Field Study Permissions

The following information was supplied relating to field study approvals (i.e., approving body and any reference numbers):

*Heptasuchus clarki* material was collected under a Bureau of Land Management permit facilitated by Dale Hansen. All specimens described here are in the collections at the University of Wyoming (BLM permit from 1977) or at the University of Texas at Austin Texas Vertebrate Paleontology Collections (BLM permit PA09-WY-177).

## Data Availability

The phylogenetic character list, phylogenetic dataset, and supplemental figures are available in the Supplemental Files.

*Heptasuchus clarki* specimens are either curated in the collections at the University of Wyoming Geological Museum (UW) or at the Texas Vertebrate Paleontology Collections, The University of Texas at Austin (TMM). All specimens are listed in the holotype or referred specimen section in the Systematic Paleontology section.

UW 11562, partial skull:
– right premaxilla (UW 11562-A)
– right maxilla (UW 11562-B)
– left maxilla (UW 11562-C)
– right jugal (UW 11562-D)
– left jugal (UW 11562-E)
– right nasal (UW 11562-F)
– right postfrontal, postorbital, partial frontal, and prefrontal (UW 11562-G)
– occiput and braincase (UW 11562-H)
– left palatine (UW 11562-K)
– pterygoid (UW 11562-L)
– pterygoid fragment (UW 11562-M)
– fragment of hyoid? (UW 11562-N)
– unidentified skull fragments (UW 11562O through -R)
– loose teeth (UW 11562-AA through -AI).

Referred material:
– quadrate head (UW 11563-AD)
– ventral condyles of left quadrate (UW 11563-AF, UW 11563-H)
– anterior cervical vertebra (UW 11562-T)
– posterior cervical centrum (UW 11564-A)
– posterior trunk vertebra (TMM 45902-2)

- neural spine of a cervical-trunk vertebra (UW 11562-CX)
- presacral neural spine (UW 11562-V)
- presacral neural spine (UW 11562-CT)
- anterior caudal vertebra (UW 11562-U)
- distal caudal vertebra (UW 11562-BW; UW 11563-A-C)
- osteoderm (TMM 45902-1)
- right partial scapula (UW 11565-E)
- right partial scapula (UW 11566-B)
- partial left coracoid (UW 11566)
- proximal portion of left humerus (UW 11565-A)
- left humerus (UW 11563-U)
- proximal portion of the radius (UW 11562-DM)
- distal portion of the radius (UW 11562-DI; UW 11562-DF)
- right ulna (UW 11562-W)
- left ulna (UW 11562-X)
- distal ends of ulnae (UW 11563-V; UW 11565-C);
- left pubis (UW 11562-Y)
- ilium fragment (UW 11563-Y)
- pubic peduncle of the right ilium (UW 11563-Z)
- left pubis (UW 11562-Y)
- proximal portion of the right ischium (UW 11564-B)
- proximal portion of a right femur (UW 11563-B)
- distal portion of the right femur (UW 11563-A)
- left tibia (UW 11562-Z)
- proximal portion of a right fibula (UW 11566-S)
- distal portion of the right fibula (UW 11566-R)
- proximal end of metatarsals (UW 11562-DH, UW 11562-DHU, UW 11562-DR)
- ungual (UW 11562-DT).

## Supplemental Information

Supplemental information for this article can be found online at http://dx.doi.org/10.7717/peerj.10101#supplemental-information.

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
