# Peer review of "The osteology and phylogenetic position of the loricatan (Archosauria: Pseudosuchia) Heptasuchus clarki, from the ?Mid-Upper Triassic, southeastern Big Horn Mountains, Central Wyoming (USA)"

_PeerJ, doi:10.7717/peerj.10101_

## Round 0.1 · original submission · Minor Revisions

The reviewers are positive about your work, but still offer some suggestions for improvements both in form and substance. Please add a scale to Fig. 2 and check that all subfigures are to scale in Fig. 6.
Please, together with your unmarked revised manuscript, provide a marked-up copy as well as a document explaining how you have addressed each of the points raised by the reviewers.

·

Basic reporting

No comment

Experimental design

No comment

Validity of the findings

No comment

Additional comments

This mauscript is very important for better understanding of archosaurs evolution and biostratigraphy. My main suugst is to add new figure with reconstruction of the skull in lateral and dorsal views.

·

Basic reporting

The manuscript covers all four of the basic reporting areas.
This paper is an important contribution to the understanding of paracrocodylomorphs archosaurs, and subsequently, the distant ancestry of crocodylians. It is nice to see a thorough description of this taxon finally! Generally speaking, the paper is well written and clear. I have annotated the word version of the paper.
The paper reflects a current understanding of this group of reptiles with relevant literature cited with very few exceptions (see line 120).
There are some grammatical and punctuation issues (e.g., see lines 70, 286, 540, 574, 586, etc.). And I have made some suggestions here and there (e.g., lines 326, 556, 595). The figures, for the most part, look fine; however, there are a few formatting errors with some of the fig. #’s (e.g., see lines 260-265).

Experimental design

Materials and methods are fine. The description is thorough and the phylogenetic analysis has been built upon previous analyses with the results congruent with current thinking.
There are a few characters that may benefit from a closer examination (e.g., lines 441, 563). One of the characters consistent in your analyses, is the "fused interdental plates" being present in Postosuchus. As I mention in a comment in your paper, can you better differentiate what you mean by this, since you can make out the separate plates in both TTU P 9000 & 9002?
Overall, the methodology and characters used in the analysis are in line with that of other researchers within the field and should be easily replicated.

Validity of the findings

As mentioned above, a few of the characters should be reassessed (e.g., lines 441, 563), but most likely this will not dramatically change the topology of the provided trees.
The conclusions follow the results of other authors and seems to place Heptasuchus in a taxonomic position that makes sense, given understanding of known taxa.
As stated by the authors, this taxon fits into a little understood gap of geologic time in North America, so any new information is welcome, and increases the significance of this paper.

Additional comments

Overall, a nice, thorough description of the known Heptasuchus material. Please see comments throughout the attached word file. I have attempted to fix some minor errors, a few of them stylistic, so take that as you will, and some suggestions to make the paper flow better. Several, just grammar and punctuation. There are some missing words and other minor things, but generally speaking, a welcome addition to the literature on paracrocodylomorph archosaurs. Would of course be nice to better constrain the time of deposition of the fossils, but it's nice to see more careful consideration of the relationship of this material to the Popo Agie Fm.

·

Basic reporting

Overall this is an important paper, it’s amazing that this fossil has escaped further serious scrutiny since its first publication 40 years ago. Overall, the paper reads well, but there are numerous minor grammatical points that require repair. I think most should be obvious from the uploaded copy of the manuscript, but feel free to contact me if not.

The single most pressing need is to universally replace the many instances of “kuperferzellensis” with “kupferzellensis,” the correct specific epithet for Batrachotomus. Both are used throughout the text.

Some other issues I would suggest:
Figure 1—the map really is an outcrop or bedrock map of the Chugwater, not a map of
“surface deposits.” Perhaps “Distribution of Chugwater Group strata.” Also, I’d like to see the age of the Gypsum Spring Fm indicated.
Figure 2 would benefit greatly from some indication of scale, which should be easy to extrapolate from Figure 4.

Generally speaking, I would like to see more call-outs to the figures in the text, including at the beginning of the Systematic Paleontology and at least once a paragraph when dealing with the individual bones, especially the various minute processes or other details that may not be labelled. Some indication is provided on the attached copy of the manuscript.

The Jelm is a nefarious beast that is remarkably understudied (words to that effect in the following), but it is a more recent and more synthetic treatment that I think is worth citing.

Blakey, R.C., Peterson, F., and Kocurek, G. 1988. Synthesis of late Paleozoic and Mesozoic eolian deposits of the Western Interior of the United States. Sedimentary Geology, 56(1-4):3-125.

For completeness’ sake the authors may wish to include Lucas (1994) for discussion of the Triassic of Wyoming.

Lucas, S.G. 1994. The beginning of the age of dinosaurs in Wyoming. Wyoming Geological Association Guidebook, 44:105-113.

References:

Somehow Gower (1999) escaped the bibliography

Gower, D.J. 1999. The cranial and mandibular osteology of a new rauisuchian archosaur from the Middle Triassic of southern Germany. Stuttgarter Beiträge zur Naturkunde Serie B (Geologie und Paläontologie), 280:1-49.

References in the text are to Bittencourt et al. (2014) but in the references is Bittencourt et al. (2015).

I don’t believe Brusatte et al. (2011) is cited in the paper, but is in the references. If cited, capitalize and italicize Ctenosauriscus.

I don’t believe Colbert (1952) is cited in the text, but is in the references.

The text refers to both Nesbitt & Desojo (2017) and 2018, but only 2017 is in the text

Line 135—High and Picard (1965) is listed as High and Picard (1969) in the references. Lines 158-159 do refer to H&P 1969, so either H&P 1965 is missing or misidentified.

Johnson (1993) does not appear to be cited and appears to be an incomplete citation of a USGS publication.

Maddison & Maddison (2015) does not appear to be cited.

Line 505—Nesbitt & Desojo 2018 is probably 2017, or else needs to be added to references.

Lines 1681 and 1722—Schoch & Desojo is 2016, not 2015

Line 992—Sill is 1974, not 1971,

Finally, within either the acknowledgments or the materials & methods the authors may wish to acknowledge the preparation methods used (and/or the preparators), as I do think that this has long-term benefits.

Experimental design

This was, of course, well-executed and probably the highlight of the paper. It is great to see this fossil documented so thoroughly and in a modern phylogenetic context. This longer contribution is the kind of thing that PeerJ is well known for and this paper is certainly a valid contribution.

I would like to see a few more of the features discussed in the text identified on the figures and, as noted earlier, some more call-outs so that the reader always knows which sub-figure is best for some of the details that are perhaps difficult to label.

Validity of the findings

Overall, I find all of this readily acceptable, it is a thorough revision that clears up a taxon about which we know too little. The necessary data are present and robust.

Two points require clarification, one minor and one major.

Minor clarification: The abstract and Line 301 indicate that there are four individuals based on ulnae, but Line 948 indicates that the MNI is 3 based on the ulnae, so this needs to be clarified.

Major clarification:

The only part of this manuscript I would like to see again is the text “The phylogenetic position of Heptasuchus clarki among archosaurs,” specifically lines 1150-1170. To me the paragraphs beginning at 1150 and 1159 both seem to be indicating a list of character states supporting a sister-group relationship between H. clarki and B. kupferzellensis. I suspect that the paragraph at 1150 is actually listing states that support inclusion of both taxa within Loricata—and I also find this a little less than convincing as H. clarki is only scored for 2 of the 6 characters listed.

I absolutely do not think that this requires re-review, but I would appreciate it if the authors could generate revised paragraphs for this section that clarify their results.

Additional comments

All told I think this is valuable contribution that really only requires a couple days' work to be publishable with minor revisions.

In addition to the (mainly) minor points listed previously, I think the following will dramatically improve the paper for relatively little effort:

(1) Please include a larger view of the reconstructed skull as part of Figure 3B. While it may be tentative, that will significantly increase the impact of the paper. I think this is valid even if it’s only slightly redrawn from Gower (1999).
(2) Figure 8 is entirely too small, I assume it will be two columns in the final.

(3) Figure 9 is also pretty small—it could easily be larger as a 1-column figure (see attached), but even if it’s landscape format, the bones could be larger.

---

## Round 0.2 · Minor Revisions

Please address the suggestions for improvement offered by Reviewer 2 and answer their few remaining questions (e.g., li 1019 of the MS).

·

Basic reporting

Generally, the paper is fine and most issues have been resolved from the initial review. There are a few mostly minor grammatical and punctuation issues, however and only a couple of questions, e.g., see line 990 regarding the auricular recess.

Experimental design

The experimental design is perfectly fine.

Validity of the findings

All good here.

Additional comments

As previously stated, the paper is a great addition to archosaur literature. It reads much better now. I found a few more grammar/punctuation things mentioned in the attached word file. Otherwise, it's a very nice paper.
I do want to let you know, due to not being able to just upload the word file with track changes, some of the minor punctuation stuff is really tough to catch, I could see a few things you missed after the initial review. Otherwise, great!

---

## Round 0.3 · accepted · Accept

Please, correct the first name of Reviewer 2 in the acknowledgments when you get a chance.